# Early Stopping for Deep Image Prior

## Abstract

Deep image prior (DIP) and its variants have shown remarkable potential to solve inverse problems in computational imaging (CI), *needing no separate training data.* Practical DIP models are often substantially overparameterized. During the learning process, these models first learn the desired visual content and then pick up potential modeling and observational noise, i.e., performing early learning then overfitting. Thus, the practicality of DIP hinges on early stopping (ES) that can capture the transition period. In this regard, most previous DIP works for CI tasks only demonstrate the potential of the models, reporting the peak performance against the ground truth but providing no clue about how to operationally obtain near-peak performance *without access to the ground truth.* In this paper, we set to break this practicality barrier of DIP, and propose an effective ES strategy that consistently detects near-peak performance across several CI tasks and DIP variants. Simply based on the running variance of DIP intermediate reconstructions, our ES method not only outpaces the existing ones—which only work in very narrow regimes, but also remains effective when combined with methods that try to mitigate overfitting.

## 1 Introduction

Inverse problems (IPs) are prevalent in computational imaging (CI), ranging from basic image denoising, super-resolution, and deblurring, to advanced 3D reconstruction and major tasks in scientific and medical imaging (Szeliski, 2022). Despite the disparate settings, all these problems take the form of recovering a visual object $\boldsymbol{x}$ from $\boldsymbol{y} = f(\boldsymbol{x})$, where $f$ models the forward process to obtain the observation $\boldsymbol{y}$. Typically, these visual IPs are underdetermined: $\boldsymbol{x}$ cannot be uniquely determined from $\boldsymbol{y}$. This is exacerbated by potential modeling (e.g., linear $f$ to approximate a nonlinear process) and observational (e.g., Gaussian or shot) noise, i.e., $\boldsymbol{y} \approx f(\boldsymbol{x})$. To overcome nonuniqueness and improve noise stability, researchers often encode a variety of problem-specific priors on $\boldsymbol{x}$ when formulating IPs.

Traditionally, IPs are phrased as regularized data fitting problems:

$$\min_{\boldsymbol{x}} \ \ell(\boldsymbol{y}, f(\boldsymbol{x})) + \lambda R(\boldsymbol{x}) \qquad \ell(\boldsymbol{y}, f(\boldsymbol{x})) : \text{data-fitting loss}, \ R(\boldsymbol{x}) : \text{regularizer} \tag{1}$$

where $\lambda$ is the regularization parameter. Here, the loss $\ell$ is often chosen according to the noise model, and the regularizer $R$ encodes priors on $\boldsymbol{x}$. The advent of deep learning (DL) has revolutionized the way IPs are solved. On the radical side, deep neural networks (DNNs) are trained to directly map any given $\boldsymbol{y}$ to an $\boldsymbol{x}$; on the mild side, pre-trained or trainable DL models are taken to replace certain nonlinear mappings in numerical algorithms for solving Eq. (1) (e.g. plug-and-play and algorithm unrolling); see recent surveys Ongie et al. (2020); Janai et al. (2020) on these developments. All of these DL-based methods rely on large training sets to adequately represent the underlying priors and/or noise distributions. **This paper concerns another family of striking ideas that require no separate training data**.

**Deep image prior (DIP)** Ulyanov et al. (2018) proposes parameterizing $\boldsymbol{x}$ as $\boldsymbol{x} = G_{\boldsymbol{\theta}}(\boldsymbol{z})$, where $G_{\boldsymbol{\theta}}$ is a trainable DNN parameterized by $\boldsymbol{\theta}$ and $\boldsymbol{z}$ is a frozen or trainable random seed. **No separate training data other than $\boldsymbol{y}$ are used!** Plugging the reparametrization into Eq. (1), we obtain

$$\min_{\boldsymbol{\theta}} \ \ell(\boldsymbol{y}, f \circ G_{\boldsymbol{\theta}}(\boldsymbol{z})) + \lambda R \circ G_{\boldsymbol{\theta}}(\boldsymbol{z}). \tag{2}$$

$G_{\boldsymbol{\theta}}$ is often "overparameterized"—containing substantially more parameters than the size of $\boldsymbol{x}$, and "structured"—e.g., consisting of convolution networks to encode structural priors in natural visual objects. The resulting optimization problem is solved via standard first-order methods for modern DL (e.g., (adaptive) gradient descent). When $\boldsymbol{x}$ has multiple components with different physical meanings, one can naturally parametrize $\boldsymbol{x}$ using multiple DNNs. This simple idea has led to surprisingly competitive results in numerous visual IPs, from low-level image denoising, super-resolution, inpainting (Ulyanov et al., 2018; Heckel & Hand, 2019; Liu et al., 2019) and blind deconvolution (Ren et al., 2020; Wang et al., 2019; Asim et al., 2020; Tran et al., 2021; Zhuang et al., 2022a), to mid-level image decomposition and fusion (Gandelsman et al., 2019; Ma et al., 2021), and to advanced CI problems (Darestani & Heckel, 2021; Hand et al., 2018; Williams et al., 2019; Yoo et al., 2021; Baguer et al., 2020; Cascarano et al., 2021; Hashimoto & Ote, 2021; Gong et al., 2022; Veen et al., 2018; Tayal et al., 2021; Zhuang et al., 2022b); see the survey Qayyum et al. (2021).

**Overfitting issue in DIP**   A critical detail that we have glossed over is **overfitting**. Since $G_{\boldsymbol{\theta}}$ is often substantially overparameterized, $G_{\boldsymbol{\theta}}(\boldsymbol{z})$ can represent arbitrary elements in the $\boldsymbol{x}$ domain. Global optimization of equation 2 would normally lead to $\boldsymbol{y} = f \circ G_{\boldsymbol{\theta}}(\boldsymbol{z})$, but $G_{\boldsymbol{\theta}}(\boldsymbol{z})$ may not reproduce $\boldsymbol{x}$, e.g., when $f$ is non-injective, or $\boldsymbol{y} \approx f(\boldsymbol{x})$ so that $G_{\boldsymbol{\theta}}(\boldsymbol{z})$ also accounts for the modeling and observational noise. Fortunately, DIP models and first-order optimization methods together offer a blessing: in practice, $G_{\boldsymbol{\theta}}(\boldsymbol{z})$ has a bias towards the desired visual content and learns it much faster than learning noise. Therefore, the quality of reconstruction climbs to a peak before the potential degradation due to noise; see Fig. 1. This "early-learning-then-overfitting" (ELTO) phenomenon has been repeatedly reported in previous works and is also supported by theories on simple $G_{\boldsymbol{\theta}}$ and linear $f$ (Heckel & Soltanolkotabi, 2020b;a). **The successes of the DIP models claimed above are conditioned on that appropriate early stopping (ES) around the performance peaks can be made**.

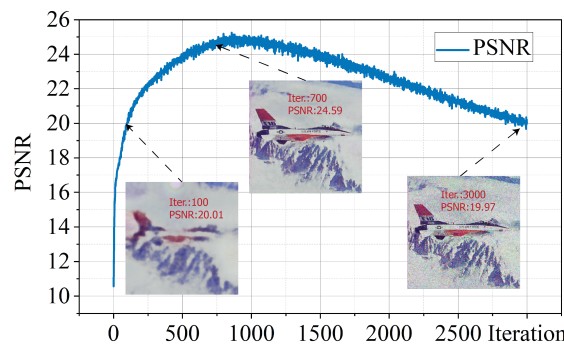

Figure 1:   The "early-learning-then-overfitting" (ELTO) phenomenon in DIP for image denoising. The quality of the estimated image climbs first to a peak and then drops once the noise is picked up by the model $G_{\boldsymbol{\theta}}(\boldsymbol{z})$ also.

**Is ES for DIP trivial?**   Natural ideas trying to perform ES can fail quickly. **(1) Visual inspection**: This subjective approach is fine for small-scale tasks involving few problem instances, but quickly becomes infeasible for many scenarios, such as (a) large-scale batch processing, (b) recovery of visual contents tricky to visualize and/or examine by eyes (e.g. 3D or 4D visual objects), and (c) scientific imaging of unfamiliar objects (e.g., MRI imaging of rare tumors and microscopic imaging of new virus species); **(2) Tracking full-reference/no-reference image quality metrics (FR/NR-IQMs) or fitting loss**: Without the ground truth $\boldsymbol{x}$, computing any FR-IQM and hence tracking their trajectories (e.g., the PNSR curve in Fig. 1) is out of the question. We consider tracking NR-IQMs as a family of baseline methods in Sec. 3.1; the performance is much worse than ours. We also explore the possibility of using the loss curve for ES here, but are unable to find correlations between the trend of the loss and that of the PSNR curve, shown in Fig. 18; **(3) Tuning the iteration number**: This ad hoc solution is taken in most previous work. But since the peak iterations of DIP vary considerably across images and tasks (see, e.g., Figs. 4 and 29 and Appendices A.7.3 and A.7.5), this could entail numerous trial-and-error steps and lead to suboptimal stopping points; **(4) Validation-based ES**: ES easily reminds us of validation-based ES in supervised learning. The DIP approach to IPs, as summarized in Eq. (2) **is not** supervised learning, as it only deals with a single instance $\boldsymbol{y}$, without separate $(\boldsymbol{x}, \boldsymbol{y})$ pairs as training data. There are recent ideas (Yaman et al., 2021; Ding et al., 2022) that hold part of the observation $\boldsymbol{y}$ out as a validation set to emulate validation-based ES in supervised learning, but they quickly become problematic for nonlinear IPs due to the significant violation of the underlying i.i.d. assumption; see Sec. 3.5.

**Prior work addressing the overfitting** There are three main approaches for countering overfitting of DIP models. **(1) Regularization**: Heckel & Hand (2019) mitigates overfitting by restricting the size of $G_{\boldsymbol{\theta}}$ to the underparameterized regime. Metzler et al. (2018); Shi et al. (2022); Jo et al. (2021); Cheng et al. (2019) control the network capacity by regularizing the norms of layer-wise weights or the network Jacobian. Liu et al. (2019); Mataev et al. (2019); Sun (2020); Cascarano et al. (2021) use additional regularizer(s) $R(G_{\boldsymbol{\theta}}(\boldsymbol{z}))$, such as the total-variation norm or trained denoisers. These methods require the right regularization level—which depends on the noise type and level—to avoid overfitting; with an improper regularization level, they can still lead to overfitting (see Fig. 4 and Sec. 3.1). Moreover, when they indeed succeed, the performance peak is postponed to the last iterations, often increasing the computational cost severalfold. **(2) Noise modeling**: You et al. (2020) models sparse additive noise as an explicit term in their optimization objective. Jo et al. (2021) designs regularizers and ES criteria specific to Gaussian and shot noise. Ding et al. (2021) explores subgradient methods with diminishing step size schedules for impulse noise with the $\ell_1$ loss, with preliminary success. These methods do not work beyond the types and levels of noise they target, whereas our knowledge of the noise in a given visual IP is typically limited. **(3) Early stopping (ES)**: Shi et al. (2022) tracks progress based on a ratio of no-reference blurriness and sharpness, but the criterion only works for their modified DIP models, as acknowledged by the authors. Jo et al. (2021) provides noise-specific regularizer and ES criterion, but it is not clear how to extend the method to unknown types and levels of noise. Li et al. (2021) proposes monitoring DIP reconstruction by training a coupled autoencoder. Although its performance is similar to ours, the extra autoencoder training slows down the whole process dramatically; see Sec. 3. Yaman et al. (2021); Ding et al. (2022) emulate validation-based ES in supervised learning by splitting elements of $\boldsymbol{y}$ into "training" and "validation" sets so that validation-based ES can be performed. But in IPs, especially nonlinear ones (e.g., in blind image deblurring—BID, $\boldsymbol{y} \approx \boldsymbol{k} * \boldsymbol{x}$ where $*$ is linear convolution), elements of $\boldsymbol{y}$ can be far from being i.i.d., and so validation may not work well. Moreover, holding out part of the observation in $\boldsymbol{y}$ can substantially reduce the peak performance; see Sec. 3.5.

Table 1: Summary of performance of our DIP+ES-WMV and competing methods on image denoising and blind image deblurring (BID). ✓: working reasonably well (PSNR $\geq 2dB$ less of the original DIP peak); -: not working well (PSNR $\leq 2dB$ less of the original DIP peak): N/A: not applicable (i.e., we do not perform comparison due to certain reasons). Note that DF-STE, DOP, and SB are based on modified DIP models.

| | Image denoising | | | | | | | | BID | |
| | Gaussian | | Impulse | | Speckle | | Shot | | Real world | |
| | Low | High | Low | High | Low | High | Low | High | Low | High |
|---|---|---|---|---|---|---|---|---|---|---|
| DIP+ES-WMV | ✓ | ✓ | ✓ | ✓ | ✓ | ✓ | ✓ | ✓ | ✓ | ✓ |
| DIP+NR-IQMs | - | - | - | - | - | - | - | - | N/A | N/A |
| DIP+SV-ES | ✓ | ✓ | ✓ | ✓ | ✓ | ✓ | ✓ | ✓ | N/A | N/A |
| DIP+VAL | ✓ | ✓ | ✓ | ✓ | ✓ | ✓ | ✓ | ✓ | - | - |
| DF-STE | ✓ | ✓ | N/A | N/A | N/A | N/A | ✓ | ✓ | N/A | N/A |
| DOP | N/A | N/A | ✓ | ✓ | N/A | N/A | N/A | N/A | N/A | N/A |
| SB | ✓ | ✓ | N/A | N/A | N/A | N/A | N/A | N/A | N/A | N/A |

**Our contribution** We advocate the ES approach—**the iteration process stops once a good ES point is detected**, as (1) the regularization and noise modeling approaches, even if effective, often do not improve peak performance but push it until the last iterations; there could be $\geq 10\times$ more iterations spent than climbing to the peak in the original DIP models; (2) both need deep knowledge about the noise type/level, which is practically unknown for most applications. If their key models and hyperparameters are not set appropriately, overfitting probably remains, and ES is still needed. **In this paper, we build a novel ES criterion for various DIP models simply by monitoring the trend of the running variance of the reconstruction sequence**. Our ES method is **(1) Effective**: The gap between our detected and the peak performance, i.e., the detection gap, is typically very small, as measured by standard visual quality metrics (PSNR and SSIM); **(2) Efficient**: The per-iteration overhead is a fraction—the standard version in Algorithm 1, or negligible—the variant in Algorithm 2, relative to the per-iteration cost of Eq. (2); **(3) General**: Our method works well for DIP and its variants, including sinusoidal representation

networks (Sitzmann et al., 2020, SIREN) and deep decoder (Heckel & Hand, 2019, DD), on different noisy types / levels and in 5 visual IPs, both linear and nonlinear. Furthermore, our method can help several regularization-based methods, e.g., Gaussian process-DIP (Cheng et al., 2019, GP-DIP), DIP with total variation regularization (Liu et al., 2019; Cascarano et al., 2021, DIP-TV) to perform reasonable ES when they fail to prevent overfitting; **(4) Robust**: Our method is relatively insensitive to the two hyperparameters, i.e. window size and patience number. We keep the same hyperparameters for all experiments Secs. 2 and 3 except for the ablation study (see Sec. 3.7). In contrast, the hyperparameters of most of the methods reviewed above are sensitive to the noise type/level. We summarize the performance of our DIP+ES method against competing methods in Tab. 1; we present the detailed results in Sec. 3.

## 2 Our Early-Stopping Method

**Intuition for our method**  We assume: $\boldsymbol{x}$ is the unknown ground-truth visual object of size $N$, $\{\boldsymbol{\theta}^t\}_{t\geq 1}$ is the iterate sequence and $\{\boldsymbol{x}^t\}_{t\geq 1}$ the reconstruction sequence where $\boldsymbol{x}^t \doteq G_{\boldsymbol{\theta}^t}(\boldsymbol{z})$. Since we do not know $\boldsymbol{x}$, we cannot access the PNSR or any FR-IQM curve. But we observe that (Fig. 2) generally the MSE (resp. PSNR; recall $\mathrm{PSNR}(\boldsymbol{x}^t) = 10\log_{10}\|\boldsymbol{x}\|_\infty^2/\mathrm{MSE}(\boldsymbol{x}^t)$) curve follows a U (resp. bell) shape: $\|\boldsymbol{x}^t - \boldsymbol{x}\|_F^2$ initially drops quickly to a low level and then climbs back due to the noise effect, i.e. the ELTO phenomenon in Sec. 1; we hope to detect the valley of this U-shaped MSE curve.

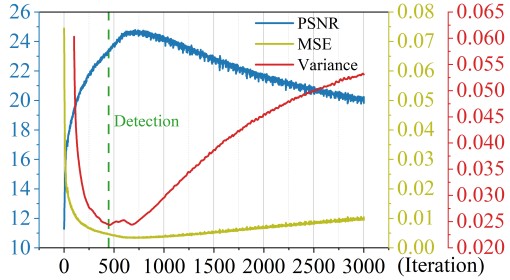

Figure 2: Relationship between the PSNR, MSE, and VAR curves. Our method relies on the VAR curve, whose valley is often well aligned with the MSE valley, to detect the MSE valley—that corresponds to the PSNR peak.

Then how to gauge the MSE curve **without knowing** $\boldsymbol{x}$? We consider the running variance (VAR):

$$\mathrm{VAR}(t) \doteq \frac{1}{W}\sum_{w=0}^{W-1}\left\|\boldsymbol{x}^{t+w} - 1/W\cdot\sum_{i=0}^{W-1}\boldsymbol{x}^{t+i}\right\|_F^2. \tag{3}$$

Initially, the models quickly learn the desired visual content, resulting in a monotonic, rapidly decreasing MSE curve (see Fig. 2). So we expect the running variance of $\{\boldsymbol{x}^t\}_{t\geq 1}$ to also drop quickly, as shown in Fig. 2. When the iteration is near the MSE valley, all $\boldsymbol{x}^{t'}$s are near, but scattered around $\boldsymbol{x}$. So $\frac{1}{W}\sum_{i=0}^{W-1}\boldsymbol{x}^{t+i} \approx \boldsymbol{x}$ and $\mathrm{VAR}(t) \approx \frac{1}{W}\sum_{w=0}^{W-1}\|\boldsymbol{x}^{t+w} - \boldsymbol{x}\|_F^2$. Afterward, the noise effect kicks in and the MSE curve bounces back, leading to a similar bounce back in the VAR curve as the $\boldsymbol{x}^t$ sequence gradually moves away from $\boldsymbol{x}$.

This argument suggests a U-shaped VAR curve and the curve should follow the trend of the MSE curve, with approximately aligned valleys, which in turn are aligned with the PSNR peak. To quickly verify this, we randomly sample 1024 images from the RGB track of the NTIRE 2020 Real Image Denoising Challenge (Abdelhamed et al., 2020), and perform DIP-based image denoising (i.e. $\min \ell(\boldsymbol{y}, G_{\boldsymbol{\theta}}(\boldsymbol{z}))$ where $\boldsymbol{y}$ denotes the noisy

Table 2: ES-WMV (our method) on real-world image denoising for **1024 images**: mean and (std) on the images. (**D**: detected)

| $\ell$ (loss) | PSNR (**D**) | PSNR Gap | SSIM (**D**) | SSIM Gap |
|---|---|---|---|---|
| MSE | 34.04 (3.68) | 0.92 (0.83) | 0.92 (0.07) | 0.02 (0.04) |
| $\ell_1$ | 33.92 (4.34) | 0.92 (0.59) | 0.93 (0.05) | 0.02 (0.02) |
| Huber | 33.72 (3.86) | 0.95 (0.73) | 0.92 (0.06) | 0.02 (0.03) |

image). Tab. 2 reports the average detected PSNR/SSIM and the average detection gaps based on our ES method (see Algorithm 1) which tries to detect the valley of the VAR curve. On average, the detection gaps are $\leq 0.95$ in PSNR and $\leq 0.02$ in SSIM, barely noticeable by the eyes! Furthermore, we provide histograms of the PSNR and SSIM gaps in Fig. 27. For over 95% of the images, our ES method obtains a PSNR gap less than $2dB$.

**Detecting transition by running variance**  Our lightweight method only involves computing the VAR curve and numerically detecting its valley—**the iteration stops once the valley is detected**. To obtain the curve, we set a window size parameter $W$ and compute the windowed moving variance (WMV). To robustly detect the valley, we introduce a patience number $P$ to tolerate up to $P$ consecutive steps of

variance stagnation. Obviously, the cost is dominated by the calculation of variance per step, which is $O(WN)$ ($N$ is the size of the visual object). In comparison, a typical gradient update step for solving Eq. (2) costs at least $\Omega(|\boldsymbol{\theta}|N)$, where $|\boldsymbol{\theta}|$ is the number of parameters in the DNN $G_{\boldsymbol{\theta}}$. Since $|\boldsymbol{\theta}|$ is typically much larger than $W$ (default: 100), our running VAR and detection incur very little computational overhead. Our entire algorithmic pipeline is summarized in Algorithm 1. To confirm the effectiveness, we provide qualitative samples in Figs. 3 and 4, with more quantitative results included in the experiment part (Sec. 3; see also Tab. 2). Fig. 3 shows on image denoising with different noise types/levels, our ES method can detect ES points that achieve near-peak performance. Similarly, our method remains effective in several popular DIP variants, as shown in Fig. 4. Note that although our detection for DIP-TV in Fig. 4 is a bit far from the peak in terms of iteration count (as the VAR curve is almost flat after the peak), the detection gap is still small ($\sim 1.29$dB).

**Seemingly similar ideas** Our running variance and its U-shaped curve are reminiscent of the classical U-shaped bias-variance tradeoff curve and, therefore, validation-based ES (Geman et al., 1992; Yang et al., 2020). But there are crucial differences: (1) our learning setting is not supervised; (2) the variance in supervised learning is with respect to the sample distribution, while our variance here pertains to the $\{\boldsymbol{x}^t\}_{t\geq1}$ sequence. As discussed in Sec. 1, we cannot directly apply validation-based ES, although it is possible to heuristically emulate it by splitting the elements in $\boldsymbol{y}$ (Yaman et al., 2021; Ding et al., 2022)—which might be problematic for nonlinear IPs. Another line of related ideas is the detection of variance-based online change points in time series analysis (Aminikhanghahi & Cook, 2017), where the running variance is often used to detect mean-shift assuming the means are piecewise constant. Here, the piecewise constancy assumption does not hold for our $\{\boldsymbol{x}^t\}_{t\geq1}$.

---

**Algorithm 1** DIP with ES–WMV

**Input:** random seed $\boldsymbol{z}$, randomly-initialized $\boldsymbol{\theta}^0$, window size $W$, patience $P$, empty queue $\mathcal{Q}$, iteration counter $k = 0$, VAR$_{\min} = \infty$
**Output:** reconstruction $\boldsymbol{x}^*$
1: **while** not stopped **do**
2:     update $\boldsymbol{\theta}$ via Eq. (2) to obtain $\boldsymbol{\theta}^{k+1}$ and $\boldsymbol{x}^{k+1}$
3:     push $\boldsymbol{x}^{k+1}$ to $\mathcal{Q}$, pop queue if $|\mathcal{Q}| > W$
4:     **if** $|\mathcal{Q}| = W$ **then**
5:         compute VAR of elements in $\mathcal{Q}$ via Eq. (3)
6:         **if** VAR $<$ VAR$_{\min}$ **then**
7:             VAR$_{\min} \leftarrow$ VAR, $\boldsymbol{x}^* \leftarrow \boldsymbol{x}^{k+1}$
8:         **end if**
9:         **if** VAR$_{\min}$ stagnates for $P$ iterations **then**
10:           stop and return $\boldsymbol{x}^*$
11:         **end if**
12:     **end if**
13:     $k = k + 1$
14: **end while**

---

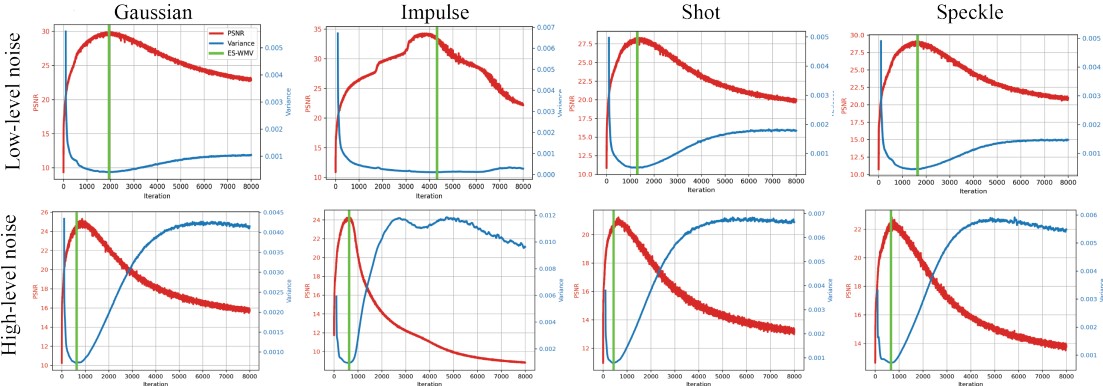

Figure 3: Our ES-WMV method on DIP for denoising "F16" with different noise types and levels (top: low-level noise; bottom: high-level noise). Red curves are PSNR curves, and blue curves are VAR curves. The green bars indicate the detected ES point.

**Theoretical justification** We can make our heuristic argument in Sec. 2 more rigorous by restricting ourselves to additive denoising, that is, $\boldsymbol{y} = \boldsymbol{x} + \boldsymbol{n}$, and appealing to the popular linearization strategy (i.e. neural tangent kernel Jacot et al. (2018); Heckel & Soltanolkotabi (2020b)) in understanding DNN. The idea is based on the assumption that during DNN training $\boldsymbol{\theta}$ does not move much away from initialization $\boldsymbol{\theta}^0$, so

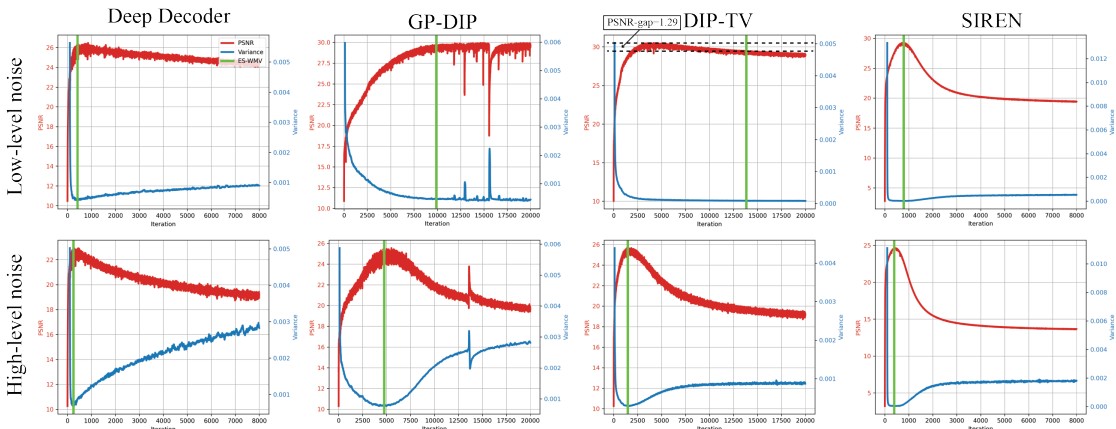

Figure 4: ES-WMV on DD, GP-DIP, DIP-TV, and SIREN for denoising "F16" with different levels of Gaussian noise (top: low-level noise; bottom: high-level noise). Red curves are PSNR curves, and blue curves are VAR curves. The green bars indicate the detected ES point. (We sketch the details of the DIP variants above in Appendix A.5)

that the learning dynamic can be approximated by that of a linearized model, i.e. suppose that we take the MSE loss,

$$\|\boldsymbol{y} - G_{\boldsymbol{\theta}}(\boldsymbol{z})\|_2^2 \approx \|\boldsymbol{y} - G_{\boldsymbol{\theta}^0}(\boldsymbol{z}) - \boldsymbol{J}_G(\boldsymbol{\theta}^0)(\boldsymbol{\theta} - \boldsymbol{\theta}^0)\|_2^2 \doteq \widehat{f}(\boldsymbol{\theta}), \tag{4}$$

where $\boldsymbol{J}_G(\boldsymbol{\theta}^0)$ is the Jacobian of $G$ with respect to $\boldsymbol{\theta}$ in $\boldsymbol{\theta}^0$, and $G_{\boldsymbol{\theta}^0}(\boldsymbol{z}) + \boldsymbol{J}_G(\boldsymbol{\theta}^0)(\boldsymbol{\theta} - \boldsymbol{\theta}^0)$ is the first-order Taylor approximation to $G_{\boldsymbol{\theta}}(\boldsymbol{z})$ around $\boldsymbol{\theta}^0$. $\widehat{f}(\boldsymbol{\theta})$ is simply a least-squares objective. We can directly calculate the running variance based on the linear model, as shown below.

**Theorem 2.1.** *Let $\sigma_i$'s and $\boldsymbol{w}_i$'s be the singular values and left singular vectors of $\boldsymbol{J}_G(\boldsymbol{\theta}^0)$, and suppose that we run a gradient descent with step size $\eta$ on the linearized objective $\widehat{f}(\boldsymbol{\theta})$ to obtain $\{\boldsymbol{\theta}^t\}$ and $\{\boldsymbol{x}^t\}$ with $\boldsymbol{x}^t \doteq G_{\boldsymbol{\theta}^0}(\boldsymbol{z}) + \boldsymbol{J}_G(\boldsymbol{\theta}^0)(\boldsymbol{\theta}^t - \boldsymbol{\theta}^0)$. Then, provided that $\eta \le 1/\max_i(\sigma_i^2)$,*

$$\mathrm{VAR}(t) = \sum_i C_{W,\eta,\sigma_i} \langle \boldsymbol{w}_i, \widehat{\boldsymbol{y}} \rangle^2 \left(1 - \eta\sigma_i^2\right)^{2t}, \tag{5}$$

*where $\widehat{\boldsymbol{y}} = \boldsymbol{y} - G_{\boldsymbol{\theta}^0}(\boldsymbol{z})$, and $C_{W,\eta,\sigma_i} \ge 0$ depend only on $W$, $\eta$, and $\sigma_i$ for all $i$.*

The proof can be found in Appendix A.2. Theorem 2.1 shows that if the learning rate (LR) $\eta$ is sufficiently small, the WMV of $\{\boldsymbol{x}^t\}$ decreases monotonically. We can develop a complementary upper bound for the WMV that has a U shape. To this end, we make use of Theorem 1 of Heckel & Soltanolkotabi (2020b), which can be summarized (some technical details omitted; precise statement is reproduced in Appendix A.3) as follows: consider the two-layer model $G_{\boldsymbol{C}}(\boldsymbol{B}) = \mathrm{ReLU}(\boldsymbol{U}\boldsymbol{B}\boldsymbol{C})\boldsymbol{v}$, where $\boldsymbol{C} \in \mathbb{R}^{n \times k}$ models $1 \times 1$ trainable convolutions, $\boldsymbol{v} \in \mathbb{R}^{k \times 1}$ contains fixed weights, $\boldsymbol{U}$ is an upsampling operation and $\boldsymbol{B}$ is the fixed random seed. Let $\boldsymbol{J}$ be a reference Jacobian matrix solely determined by the upsampling operation $\boldsymbol{U}$, and $\sigma_i$'s and $\boldsymbol{w}_i$'s the singular values and left singular vectors of $\boldsymbol{J}$. Assume $\boldsymbol{x} \in \mathrm{span}\{\boldsymbol{w}_1, \ldots, \boldsymbol{w}_p\}$. Then, when $\eta$ is sufficiently small, with high probability,

$$\|G_{\boldsymbol{C}^t}(\boldsymbol{B}) - \boldsymbol{x}\|_2 \le \left(1 - \eta\sigma_p^2\right)^t \|\boldsymbol{x}\|_2 + E(\boldsymbol{n}) + \varepsilon\|\boldsymbol{y}\|_2, \tag{6}$$

where $\varepsilon > 0$ is a small scalar related to the structure of the network and $E(\boldsymbol{n})$ is the error introduced by noise: $E^2(\boldsymbol{n}) \doteq \sum_{j=1}^{n}((1 - \eta\sigma_j^2)^t - 1)^2\langle \boldsymbol{w}_j, \boldsymbol{n}\rangle^2$. So, if the gap $\sigma_p/\sigma_{p+1} > 1$, $\|G_{\boldsymbol{C}^t}(\boldsymbol{B}) - \boldsymbol{x}\|_2$ is dominated by $\left(1 - \eta\sigma_p^2\right)^t \|\boldsymbol{x}\|_2$ when $t$ is small and then by $E(\boldsymbol{n})$ when $t$ is large. However, since the former decreases and the latter increases as $t$ grows, the upper bound has a U shape with respect to $t$. On the basis of this result, we have the following.

**Theorem 2.2.** *Assume the same setting as Theorem 2 of Heckel & Soltanolkotabi (2020b). With high probability, our WMV is upper bounded by*

$$\frac{12}{W}\|\boldsymbol{x}\|_2^2 \frac{\left(1-\eta\sigma_p^2\right)^{2t}}{1-(1-\eta\sigma_p^2)^2} + 12\sum_{i=1}^n \left(\left(1-\eta\sigma_i^2\right)^{t+W-1}-1\right)^2(\boldsymbol{w}_i^\mathsf{T}\boldsymbol{n})^2 + 12\varepsilon^2\|\boldsymbol{y}\|_2^2. \tag{7}$$

The exact statement and proof can be found in Appendix A.3. Using a reasoning similar to above, we can conclude that the upper bound in Theorem 2.2 also has a U shape. To interpret the results, Fig. 5 shows the curves (as functions of $t$) predicted by Theorems 2.1 and 2.2. The actual VAR curve should lie between the two curves. These results are primitive and limited, similar to the situations for many DL theories that provide loose upper and lower bounds; we leave a complete theoretical justification for future work.

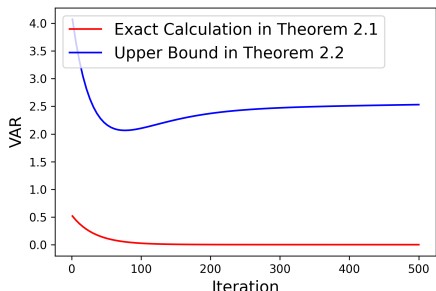

Figure 5: The exact and upper bounds predicted by Theorems 2.1 and 2.2.

**A memory-efficient variant** While Algorithm 1 is already lightweight and effective in practice, we can modify it slightly to avoid maintaining $\mathcal{Q}$ and therefore saving memory. The trick is to use exponential moving variance (EMV), together with exponential moving average (EMA), shown in Appendix A.4. The hard window size parameter $W$ is now replaced by the soft forgetting factor $\alpha$: the larger the $\alpha$, the smaller the impact of the history, and hence a smaller effective window. We systematically compare ES-WMV with ES-EMV in Appendix A.7.9 for image denoising tasks. The latter has slightly better detection due to the strong smoothing effect ($\alpha = 0.1$). For this paper, we prefer to remain simple and leave systematic evaluations of ES-EMV on other IPs for future work.

## 3 Experiments

We test ES-WMV for DIP in **image denoising, inpainting, super-resolution, MRI reconstruction, and blind image deblurring**, spanning both linear and nonlinear IPs. For image denoising, we also systematically evaluate ES-WMV in the major variants of DIP, including DD (Heckel & Hand, 2019), DIP-TV (Cascarano et al., 2021), GP-DIP (Cheng et al., 2019), and demonstrate ES-WMV as a reliable helper in detecting good ES points. Details of the DIP variants are discussed in Appendix A.5. We also compare ES-WMV with the main competing methods, including DF-STE (Jo et al., 2021), SV-ES (Li et al., 2021), DOP (You et al., 2020), SB (Shi et al., 2022), and VAL (Yaman et al., 2021; Ding et al., 2022). Details of the main ES-based methods can be found in Appendix A.6. We use both PSNR and SSIM to assess the reconstruction quality and report PSNR and SSIM gaps (the difference between our de-

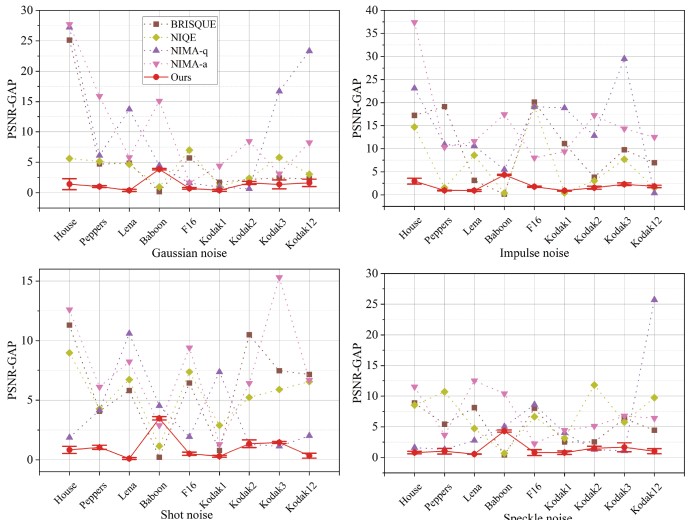

Figure 6: Baseline ES vs our ES-WMV on denoising with **low-level noise**. For NIMA, we report both technical quality assessment (NIMA-q) and aesthetic assessment (NIMA-a). Smaller PSNR gaps are better.

tected and peak numbers) as indicators of our detection performance. **Common acronyms, pointers to external codes, detailed experiment settings, results on real-world denoising are in Appendices A.1, A.7.1, A.7.2 and A.7.7, respectively.**

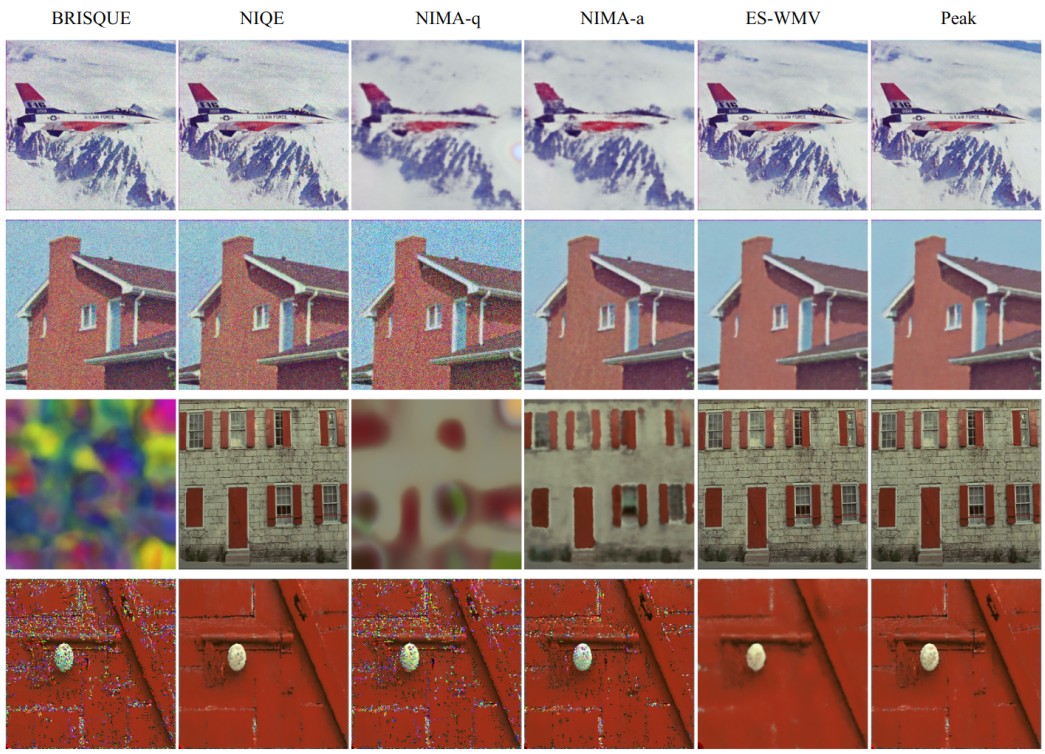

Figure 7: Visual comparisons of NR-IQMs and ES-WMV. From top to bottom: Gaussian noise (low), Gaussian noise (high), impulse noise (low), impulse noise (high).

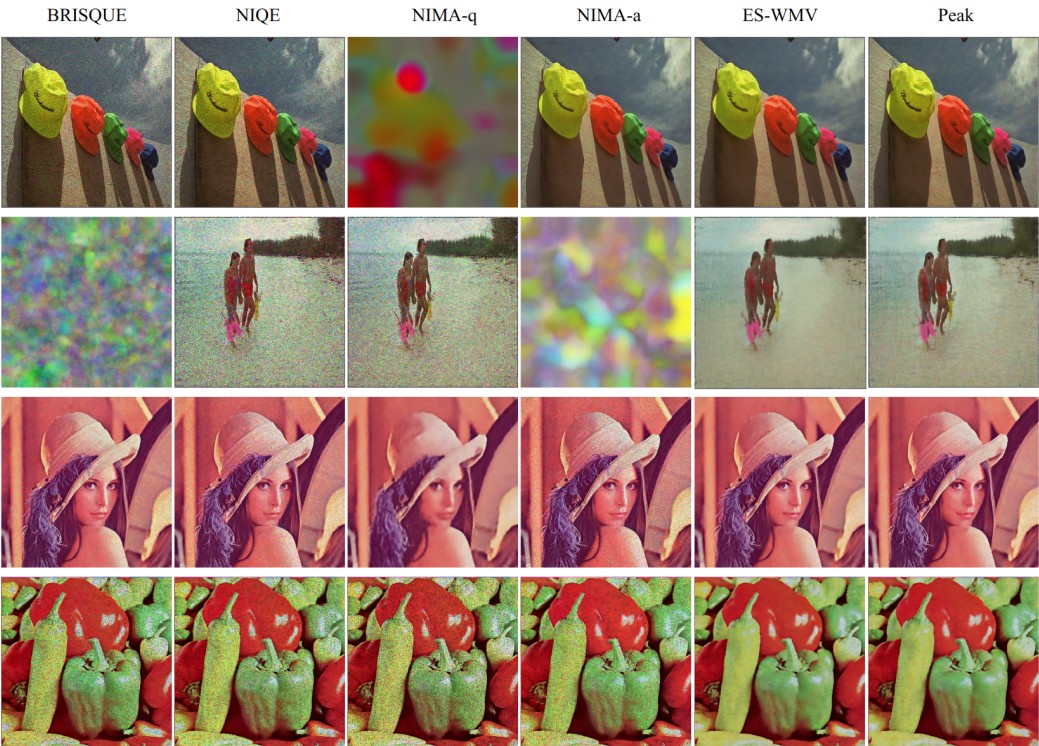

Figure 8: Visual comparisons of NR-IQMs and ES-WMV. From top to bottom: shot noise (low), shot noise (high), speckle noise (low), speckle noise (high).

### 3.1 Image denoising

Prior work dealing with DIP overfitting mostly focuses on image denoising but typically only evaluates their methods on one or two kinds of noise with low noise levels, e.g., low-level Gaussian noise. To stretch our evaluation, we consider 4 types of noise: Gaussian, shot, impulse, and speckle. We take the classical 9-image dataset (Dabov et al., 2008), and for each noise type, generate two noise levels, low and high, i.e., level 2 and 4 of Hendrycks & Dietterich (2019), respectively. See also Tab. 2 and Appendix A.7.7 about the performance of our ES-WMV on real-world denoising evaluated on **large-scale datasets**.

**Comparison with baseline ES methods** It is natural to expect that NR-IQMs, such as the classical BRISQUE (Mittal et al., 2012), NIQE (Mittal et al., 2013) and modern DNN-based NIMA (Esfandarani & Milanfar, 2018), can be used to monitor the quality of intermediate reconstructions and hence induce natural ES criteria. Therefore, we set 3 baseline methods using BRISQUE, NIQE, and NIMA, respectively, and seek the optimal $x^t$ using these metrics. Fig. 6 presents the comparison (in terms of PSNR gaps) of these 3 methods with our ES-WMV on denoising with low-level noise; results on high-level noise and also as measured by SSIM are included in Appendix A.7.4. Visual comparisons between our ES-WMV and the baseline methods are shown in Figs. 7 and 8. While **our method enjoys favorable detection gaps** ($\leq 2$) for most tested noise types/levels (except for Baboon, Kodak1, Kodak2 for certain noise types/levels; DIP itself is suboptimal in terms of denoising such images with substantial high-frequency components), **detection gaps by the baseline methods can get huge ($\geq 10$).**

**Competing methods** DF-STE (Jo et al., 2021) is specific for Gaussian and Poisson denoising, and noise variance is needed for their tuning parameters. Fig. 9 presents the comparison with DF-STE in terms of PSNR; SSIM results are in Appendix A.7.5. Here, we directly report the final PSNRs obtained by both methods. For low-level noise, there is no clear winner. **For high-level noise, ES-WMV outperforms DF-STE by considerable margins.** Although the right variance

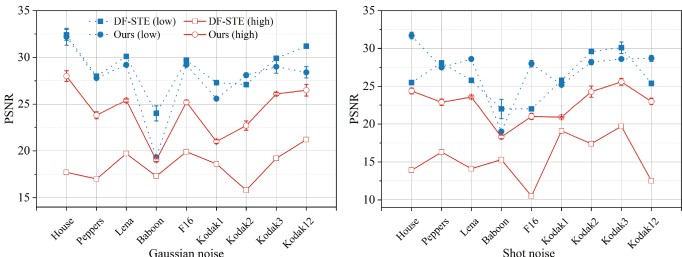

Figure 9: Comparison of DF-STE and ES-WMV for Gaussian and shot noise in terms of PSNR.

level is provided to DF-STE in order to tune their regularization parameters, DF-STE stops after only very few epochs, leading to very low performance and almost zero standard deviations—they return almost the noisy input. However, we do not perform any parameter tuning for ES-WMV. Furthermore, we compare the two methods on the CBSD68 dataset in Appendix A.7.5 with a similar conclusion.

We report the results of SV-ES in Appendix A.7.5 since ES-WMV performs largely comparable to SV-ES. However, ES-WMV is much faster in wall-clock time, as reported in Tab. 3: for each epoch, the overhead of our ES-WMV is less than 3/4 of the DIP update itself, while SV-ES is around 25× of that. There is no surprise: while our method only needs to update the running variance of $\{x^t\}_{t \geq 1}$ each time, **SV-ES needs to train a coupled autoencoder which is extremely expensive.**

Table 3: Wall-clock time (secs) of DIP and three ES methods per epoch on *NVIDIA Tesla K40 GPU*: mean and (std). The total wall clock time should contain both DIP and a certain ES method.

| DIP | SV-ES | ES-WMV | ES-EMV |
|---|---|---|---|
| Time  0.448 (0.030) | **13.027 (3.872)** | 0.301 (0.016) | 0.003 (0.003) |

DOP is **designed specifically just for impulse noise**, so we compare ES-WMV with DOP on impulse noise (see Appendix A.7.5). The loss is changed to $\ell_1$ to account for the sparse noise. In terms of the final PSNRs, DOP outperforms DIP with ES-WMV by a small gap, but even the peak PSNR of DIP with $\ell_1$ lags behind DOP by about 2dB for high noise levels.

Table 4: Comparison between ES-WMV and SB for image denoising on the CBSD68 dataset with varying noise level $\sigma$. The higher PSNR detected and earlier detection are better, which are in red: mean and (std).

| | $\sigma = 15$ | | $\sigma = 25$ | | $\sigma = 50$ | |
|---|---|---|---|---|---|---|
| | PSNR | Epoch | PSNR | Epoch | PSNR | Epoch |
| WMV | 28.7(3.2) | 3962(2506) | 27.4(2.6) | 3068(2150) | 24.2(2.3) | 1548(1939) |
| SB | 29.0(3.1) | 4908(1757) | 27.3(2.2) | 5099(1776) | 23.0(1.0) | 5765(1346) |

**The ES method in SB is acknowledged to fail for vanilla DIP (Shi et al., 2022).** Moreover, their modified model still suffers from the overfitting issue beyond the very low noise levels, as shown in Fig. 24. Their ES method fails to stop at appropriate places when the noise level is high. Hence, we test both ES-WMV and SB on their modified DIP model in (Shi et al., 2022), based on the two datasets they test: the classic 9-image dataset (Dabov et al., 2008) and the CBSD68 dataset (Martin et al., 2001). The qualitative results on the 9 images are shown in Appendix A.7.5; detected PSNR and stop epochs on the CBSD68 dataset are reported in Tab. 4. For SB, the detection threshold parameter is set to 0.01. It is evident that both methods have similar detection performance for low noise levels, but ES-WMV outperforms SB when the noise level is high. Also, ES-WMV tends to stop much earlier than SB, saving computational cost.

We compare VAL with our ES-WMV on the 9-image dataset with low-/high-level Gaussian and impulse noise. Since Ding et al. (2022) takes 90% pixels to train DIP and this usually decreases the peak performance, we report the final PSNRs detected by both methods (see Fig. 10). The two ES methods **perform very comparably in image denoising**, which is probably due to a mild violation of the i.i.d. assumption only, and also to a relatively low degree of information loss due to data splitting. **The more complex nonlinear BID in Sec. 3.5 reveals their gap.**

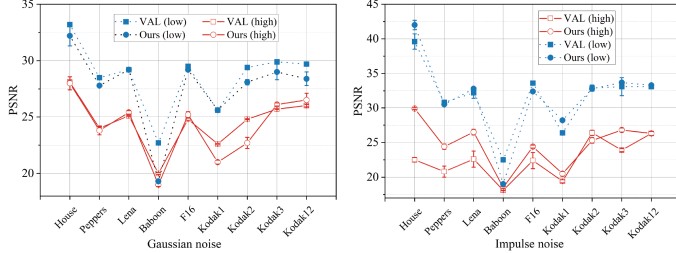

Figure 10: Comparison of VAL and ES-WMV for Gaussian and impulse noise in terms of PSNR.

**ES-WMV as a helper for DIP variants** DD, DIP-TV, and GP-DIP represent different regularization strategies to control overfitting. However, a critical issue is setting the right hyperparameters for them so that overfitting is removed while peak-level performance is preserved. Therefore, practically, these methods are not free from overfitting, especially when the noise level is high. Thus, instead of treating them as competitors, we test whether ES-WMV can reliably detect good ES points for them. We focus on Gaussian denoising and report the results in Fig. 11 (a)-(c) and Appendix A.7.6. **ES-WMV is able to attain $\leq 1$ PNSR gap for most of the cases**, with few outliers; we provide a detailed analysis about some of the outliers in Sec. 3.6.

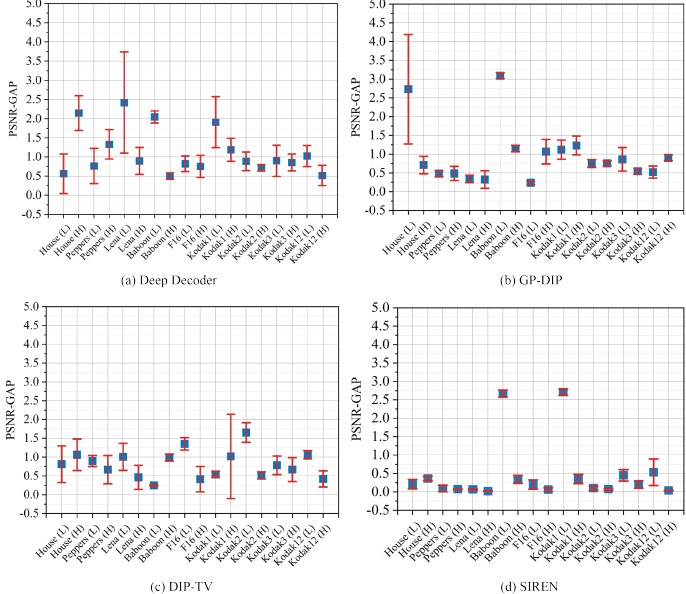

Figure 11: Performance of ES-WMV on DD, GP-DIP, DIP-TV, and SIREN for Gaussian denoising in terms of PSNR gaps. L: low noise level; H: high noise level.

**ES-WMV as a helper for implicit neural representations (INRs)** INRs, such as Tancik et al. (2020) and Sitzmann et al. (2020), use multilayer perceptrons to represent highly nonlinear functions in low-dimensional problem domains and have achieved superior results on complex 3D visual tasks. We further extend our ES-WMV to help the INR family and take SIREN (Sitzmann et al., 2020) as an example. SIREN parameterizes $x$ as the discretization of a continuous function: this function takes in spatial coordinates and returns the corresponding function values. Here, we test SIREN, which is reviewed in Appendix A.5, as a replacement of DIP models for Gaussian denoising, and summarize the results in Fig. 11 and Fig. 25. **ES-WMV is again able to detect near-peak performance for most images.**

## 3.2  Image Inpainting

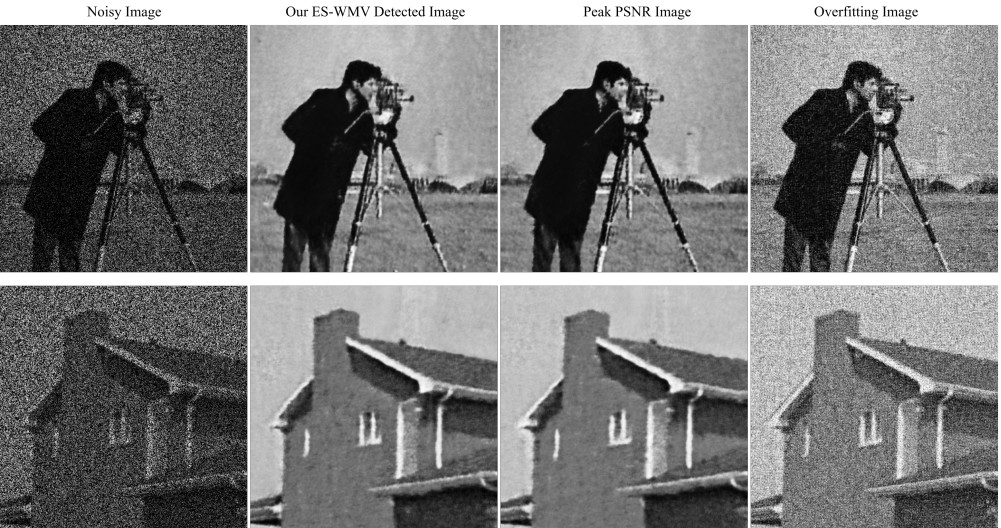

| Noisy Image | Our ES-WMV Detected Image | Peak PSNR Image | Overfitting Image |

Figure 12: Visual detection performance of ES-WMV on image inpainting.

In this task, a clean image $\boldsymbol{x}_0 \in [0,1]^{H \times W}$ is contaminated by additive Gaussian noise $\varepsilon$, and then only partially observed to yield the observation $\boldsymbol{y} = (\boldsymbol{x}_0 + \boldsymbol{\varepsilon}) \odot \boldsymbol{m}$, where $\boldsymbol{m} \in \{0,1\}^{H \times W}$ is a binary mask and $\odot$ denotes the Hadamard product. Given $\boldsymbol{y}$ and $\boldsymbol{m}$, the goal is to reconstruct $\boldsymbol{x}_0$. We consider the formulation reparametrized by DIP, where $G_{\boldsymbol{\theta}}$ is a trainable DNN parametrized by $\boldsymbol{\theta}$ and $\boldsymbol{z}$ is a frozen random seed:

$$\ell(\boldsymbol{\theta}) = \|(G_{\boldsymbol{\theta}}(\boldsymbol{z}) - \boldsymbol{y}) \odot \boldsymbol{m}\|_F^2. \tag{8}$$

Mask $\boldsymbol{m}$ is generated according to an i.i.d. Bernoulli model with a rate of 50%, i.e., half of pixels not observed in expectation. The **noise $\varepsilon$ is set to the medium level**, i.e., additive Gaussian with 0 mean and 0.18 variance. We test our ES-WMV for DIP on the inpainting dataset used in the original DIP paper Ulyanov et al. (2018). PSNR gaps are $\leq 1.00$ and SSIM gaps are $\leq 0.05$ for most cases (see Tab. 10). We also visualize two examples in Fig. 12.

## 3.3  Image Super-Resolution

In this task, a degraded observation $\boldsymbol{y}$ is obtained as the downsampled version of a noisy image: $\boldsymbol{y} = \mathcal{D}_t(\boldsymbol{x}_0 + \boldsymbol{\varepsilon})$, where $\mathcal{D}_t(\cdot) : [0,1]^{3 \times tH \times tW} \to [0,1]^{3 \times H \times W}$ is a *downsampling operator* that resizes an image by the factor $t$. Then given $\boldsymbol{y}$ and $t$, the goal is to reconstruct $\boldsymbol{x}_0$. We consider the formulation reparameterized by DIP, where $G_{\boldsymbol{\theta}}$ is a trainable DNN parameterized by $\boldsymbol{\theta}$ and $\boldsymbol{z}$ is a frozen random seed:

$$\ell(\boldsymbol{\theta}) = \|\mathcal{D}_t(G_{\boldsymbol{\theta}}(\boldsymbol{z})) - \boldsymbol{y}\|_F^2. \tag{9}$$

The **noise $\varepsilon$ is again set to the medium level**, i.e., additive Gaussian with 0 mean and 0.18 variance. We test our ES-WMV for DIP on the super-resolution dataset used in the original DIP paper Ulyanov et al. (2018). The PSNR gaps are $\leq 1.00$ and the SSIM gaps are $\leq 0.05$ for most cases (see Tab. 5). Our ES-WMV is again able to detect near-peak performance for most images.

## 3.4  MRI reconstruction

We further test ES-WMV on MRI reconstruction, a classical linear IP with a nontrivial forward mapping: $\boldsymbol{y} \approx \mathcal{F}(x)$, where $\mathcal{F}$ is the subsampled Fourier operator, and we use $\approx$ to indicate that the noise encountered in practical MRI imaging may be hybrid (e.g., additive, shot) and

Table 6: ConvDecoder on MRI reconstruction for **30 cases**: mean and (std). (**D**: Detected)

| PSNR(**D**) | PSNR Gap | SSIM(**D**) | SSIM Gap |
|---|---|---|---|
| 32.63 (2.36) | 0.23 (0.32) | 0.81 (0.09) | 0.01 (0.01) |

Table 5: Detection performance of DIP with ES-WMV for **4× image super-resolution**: mean and (std). PSNR gaps below 1.00 are colored as red; SSIM gaps below 0.05 are colored as blue. (**D**: Detected)

|  | PSNR(**D**) | PSNR Gap | SSIM(**D**) | SSIM Gap |
|---|---|---|---|---|
| Baboon | 17.82 (0.02) | 0.10 (0.04) | 0.38 (0.00) | 0.01 (0.01) |
| Barbara | 19.93 (0.05) | 0.04 (0.01) | 0.59 (0.01) | 0.01 (0.00) |
| Bridge | 18.04 (0.04) | 0.33 (0.09) | 0.43 (0.00) | 0.00 (0.00) |
| Coastguard | 20.76 (0.05) | 0.17 (0.13) | 0.53 (0.01) | 0.02 (0.01) |
| Comic | 16.70 (0.07) | 0.06 (0.06) | 0.45 (0.01) | 0.00 (0.00) |
| Face | 21.67 (0.12) | 0.63 (0.12) | 0.56 (0.01) | 0.06 (0.01) |
| Flowers | 18.96 (0.08) | 0.12 (0.03) | 0.56 (0.01) | 0.02 (0.00) |
| Foreman | 20.62 (0.04) | 0.35 (0.07) | 0.69 (0.00) | 0.06 (0.00) |
| Lena | 22.40 (0.07) | 0.30 (0.08) | 0.70 (0.00) | 0.04 (0.00) |
| Man | 19.94 (0.07) | 0.22 (0.05) | 0.52 (0.00) | 0.02 (0.01) |
| Monarch | 19.68 (0.90) | 1.40 (0.90) | 0.72 (0.00) | 0.03 (0.00) |
| Pepper | 21.20 (0.14) | 0.14 (0.04) | 0.67 (0.01) | 0.04 (0.01) |
| Ppt3 | 17.55 (0.10) | 0.19 (0.10) | 0.71 (0.01) | 0.01 (0.00) |
| Zebra | 19.09 (0.08) | 0.10 (0.05) | 0.56 (0.01) | 0.01 (0.01) |

uncertain. Here, we take the 8-fold undersampling and parameterize $\boldsymbol{x}$ using "Conv-Decoder" (Darestani & Heckel, 2021), a variant of DD. Due to the heavy over-parameterization, overfitting occurs and ES is needed. Darestani & Heckel (2021) directly sets the stopping point at the 2500-th epoch, and we run our ES-WMV. We visualize the performance on two random cases (C1: 1001339 and C2: 1000190 sampled from Darestani & Heckel (2021), part of the fastMRI datatset (Zbontar et al., 2018)) in Fig. 29 (quality measured in SSIM, consistent with Darestani & Heckel (2021)). It is clear that ES-WMV detects near-peak performance for both cases and is adaptive enough to yield comparable or better ES points than heuristically fixed ES points. Furthermore, we test our ES-WMV on ConvDecoder for **30 cases** from the fastMRI dataset (see Tab. 6), which **shows the precise and stable detection of ES-WMV**.

### 3.5 Blind image deblurring (BID)

In BID, a blurry and noisy image is given, and the goal is to recover a sharp and clean image. The blur is mostly caused by motion and/or optical non-ideality in the camera, and the forward process is often modeled as $\boldsymbol{y} = \boldsymbol{k} * \boldsymbol{x} + \boldsymbol{n}$, where $\boldsymbol{k}$ is the blur kernel, $\boldsymbol{n}$ models additive sensory noise, and $*$ is linear convolution to model the spatial uniformity of the blur effect (Szeliski, 2022). BID is a very challenging visual IP due to bilinearity: $(\boldsymbol{k}, \boldsymbol{x}) \mapsto \boldsymbol{k} * \boldsymbol{x}$. Recently, Ren et al. (2020); Wang et al. (2019); Asim et al. (2020); Tran et al. (2021) have tried to use DIP models to solve BID by modeling $\boldsymbol{k}$ and $\boldsymbol{x}$ as two separate DNNs, i.e., $\min_{\boldsymbol{\theta}_k, \boldsymbol{\theta}_x} \|\boldsymbol{y} - G_{\boldsymbol{\theta}_k}(\boldsymbol{z}_k) * G_{\boldsymbol{\theta}_x}(\boldsymbol{z}_x)\|_2^2 + \lambda\|\nabla G_{\boldsymbol{\theta}_x}(\boldsymbol{z}_x)\|_1 / \|\nabla G_{\boldsymbol{\theta}_x}(\boldsymbol{z}_x)\|_2$, where the regularizer is to promote sparsity in the gradient domain for the reconstruction of $\boldsymbol{x}$, as standard in BID.

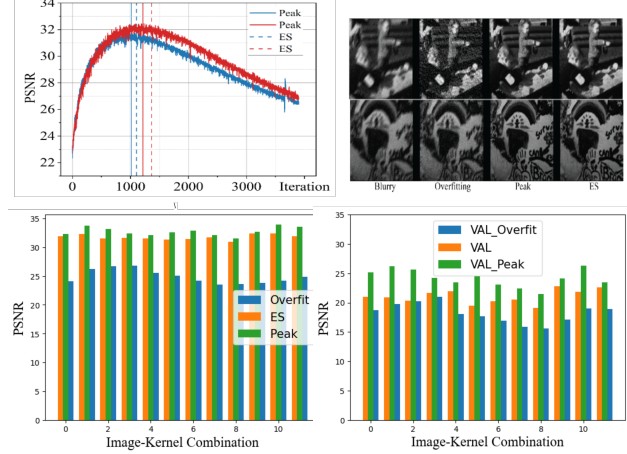

Figure 13: Top left: ES-WMV in BID; top right: visual results of ES-WMV; bottom: quantitative results of ES-WMV and VAL, respectively

We follow Ren et al. (2020) and choose multilayer perceptron (MLP) with softmax activation for $G_{\boldsymbol{\theta}_k}$, and the canonical DIP model (CNN-based encoder-decoder architecture) for $G_{\boldsymbol{\theta}_x}(\boldsymbol{z}_x)$. We change their regularizer from the original $\|\nabla G_{\boldsymbol{\theta}_x}(\boldsymbol{z}_x)\|_1$ to the current, as their original formulation is tested only on a very low noise level $\sigma = 10^{-5}$ and no overfitting is observed. We set the test with a higher noise level $\sigma = 10^{-3}$, and find

that its original formulation does not work. The benefit of the modified regularizer on BID is discussed in Krishnan et al. (2011). First, we take 4 images and 3 kernels from the standard Levin dataset (Levin et al., 2011), resulting in 12 image-kernel combinations. The high noise level leads to substantial overfitting, as shown in Fig. 13 (top left). However, ES-WMV can reliably detect good ES points and lead to impressive visual reconstructions (see Fig. 13 (top right)). We systematically compare VAL and our ES-WMV on this difficult nonlinear IP, as we suspect that nonlinearity can break down VAL as discussed in Sec. 1, and subsampling the observation $\boldsymbol{y}$ for training-validation splitting may be unwise. Our results (Fig. 13 (bottom left/right)) confirm these predictions: **the peak performance detected by VAL is much worse after** 10% **of elements in** $\boldsymbol{y}$ **are removed for valiation**. In contrast, our ES-WMV returns quantitatively near-peak performance, much better than leaving the process to overfit. In Tab. 13, we further test both low- and high-level noise on the entire Levin dataset for completeness.

### 3.6 Analysis of failure cases

Our ES method needs three things to succeed: (1) the U-shape of the VAR curve, (2) the VAR valley aligning with the PSNR peak, and (3) the successful numerical detection of the VAR valley. In this section, we discuss two major failure modes of our ES method: (I) the VAR valley aligns well with the PSNR peak, but the U-shape assumption is violated. A dominant pattern is the presence of multiple valleys, see, e.g., the top row of Fig. 14 that shows such examples with DIP variants, DD and GP-DIP (we do not observe the multi-valley phenomenon on DIP itself in Fig. 3). Since our numerical valley detection method aims to locate the first major valley, it may not locate the deepest valley among the multiple valleys. Fortunately, for these cases, we observe that using smaller learning rates (LR) can help to smooth out their curves and mitigate the multi-valley phenomenon, leading to much smaller detection gaps (see the bottom row of Fig. 14); (II) the VAR valley does not align well with the PSNR peak, which

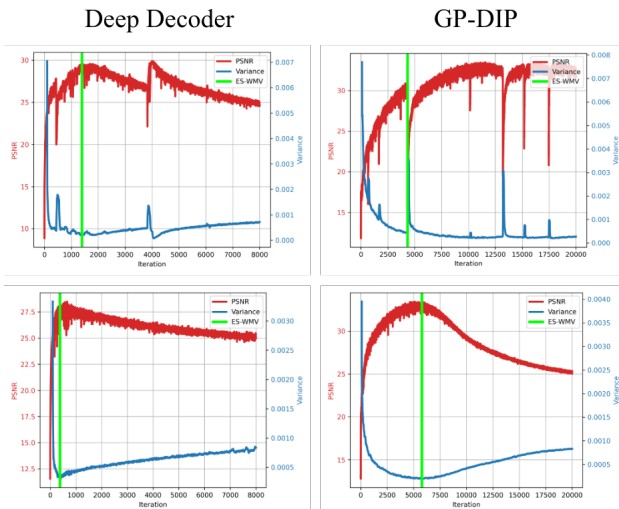

Figure 14: Top left: DD with the default LR for "Lena(L)"; top right: GP-DIP with the default LR for "House(L)"; bottom left: DD with LR = 0.001 for "Lena(L)"; bottom right: GP-DIP with LR = 0.001 for "House(L)".

often happens on images with significant high-frequency components, e.g., Fig. 30. We suspect that this is because the initial VAR decrease tends to correlate with the early learning of low-frequency components in DIP. When there are substantial high-frequency components in an image, the PSNR curve takes more time to pick up the high-frequency components, after the VAR curve already reaches the first major valley; hence, the misalignment between the VAR valley and the PSNR peak occurs.

### 3.7 Ablation study

The window size $W$ (default: 100) and the patience number $P$ (default: 1000) are the only hyperparameters for ES-WMV. To study their impact on ES detection, we vary them across a range and check how the detection gap changes for Gaussian denoising on the classic 9-image dataset (Dabov et al., 2008) with medium-level noise, as shown in Fig. 15 for PSNR gaps and Fig. 31 for SSIM gaps. Our method is robust to these changes, and it appears that larger $W$ and $P$ can produce a marginal improvement.

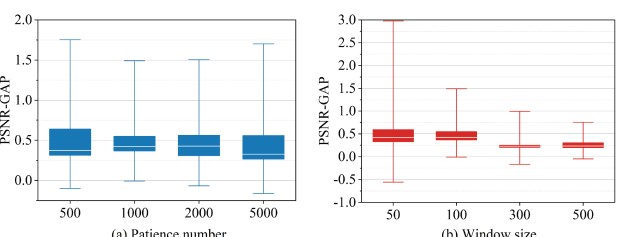

Figure 15: Effect of $W$ and $P$

In addition to the two hyperparameters of our ES-WMV, we notice that smaller learning rates (LR) can smooth out the VAR curves and mitigate the multi-valley phenomenon in Fig. 14. Therefore, we apply our

ES-WMV to DD and GP-DIP with smaller LRs (both 0.001), as shown in Fig. 16. Compared to the results of DD and GP-DIP with the default LRs in Fig. 11, most of the PSNR gaps decrease.

### 3.8 Potential for effective ES in zero-shot learning with diffusion models

Recently, zero-shot diffusion-model-based (ZS-DMB) methods have been proposed to solve linear image restoration tasks Wang et al. (2022)[1]. However, these methods rely on pre-trained models from large training datasets, while DIP does not need any training data or pre-trained models. Hence, for the sake of fairness, we do not compare DIP-based methods with ZSDMB methods in this paper. But we stress that ZSDMB methods for inverse problems may also have similar overfitting issues to those in DIP methods, especially when the observation $\boldsymbol{y}$ is noisy: as shown in Fig. 17, the overfitting issue is evident with or without additional noise. Interestingly, our ES method

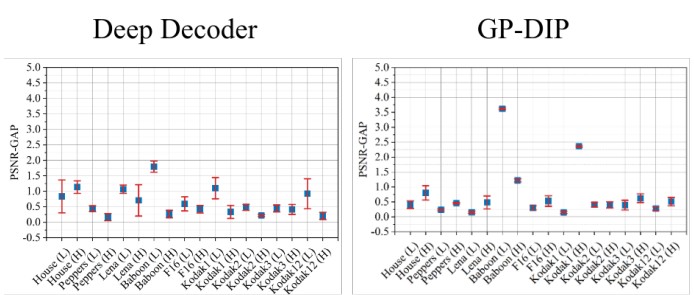

Figure 16: Performance of ES-WMV on DD and GP-DIP with smaller LRs for Gaussian denoising in terms of PSNR gaps. L: low noise level; H: high noise level.

can also help them to detect near-peak performance! We emphasize that the experiment here is exploratory and preliminary, and tackling the overfitting issue in ZSDMB methods is out of the scope of this paper. We leave a complete study for future work.

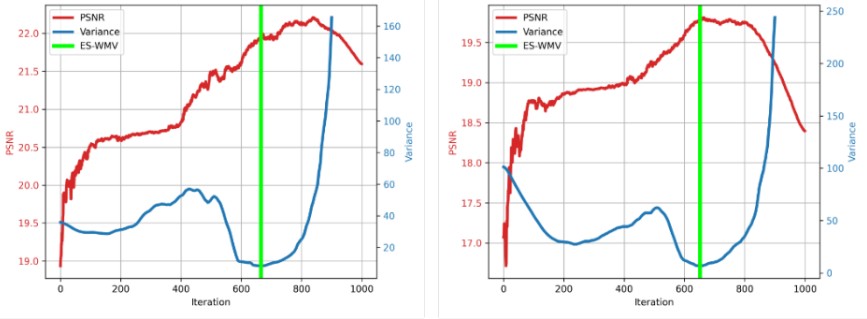

Figure 17: PSNR and VAR curves of **a zero-shot diffusion-model-based method** for **4×** image super-resolution task. The original paper Wang et al. (2022) fixes the iteration number as 1000. Left: super-resolution for clean "Butterfly"; right: super-resolution for "Butterfly" with high-level Gaussian noise.

## 4 Discussion

We have proposed a simple yet effective ES detection method (ES-WMV, and the ES-EMV variant) that works robustly across multiple visual IPs and DIP variants. In comparison, most competing ES methods are noise or DIP-model specific and only work for limited scenarios; Li et al. (2021) has comparable performance but slows down the running speed too much; validation-based ES (Ding et al., 2022) works well for the simple denoising task while significantly lags behind our ES method on nonlinear IPs, e.g. BID. As for limitations, our theoretical justification is only partial, sharing the same difficulty of analyzing DNNs in general; our ES method struggles with images with substantial high-frequency components; our detection is sometimes off the peak in terms of iteration numbers when helping certain DIP variants, e.g. DIP-TV with low-level Gaussian noise (Fig. 4), but the detected PSNR gap is still small. DIP variants typically do not improve peak performance and also do not necessarily avoid overfitting, especially for high-level noise. We recommend the

---

[1]https://github.com/wyhuai/DDNM/tree/main/hq_demo

original DIP with our ES method for visual IPs discussed in this paper for the best performance and overall speed.

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

## A  Appendix

### A.1  Acronyms

| **List of Common Acronyms** (in alphabetic order) | |
| --- | --- |
| CI | computational imaging |
| CNN | convolutional neural network |
| DD | deep decoder |
| DIP | deep image prior |
| DIP-TV | DIP with total variation regularization |
| DL | deep learning |
| DNN | deep neural network |
| ELTO | early-learning-then-overfitting |
| ES | early stopping |
| EMA | exponential moving average |
| EMV | exponential moving variance |
| FR-IQM | full-reference image quality metric |
| GP-DIP | Gaussian process DIP |
| INR | implicit neural representations |
| IP | inverse problem |
| MSE | mean squared error |
| NR-IQM | no-reference image quality metric |
| PSNR | peak signal-to-noise ratio |
| SIREN | sinusoidal representation networks |
| SOTA | state-of-the-art |
| VAR | variance |
| WMV | windowed moving variance |

## A.2 Proof of 2.1

*Proof.* To simplify the notation, we write $\widehat{\boldsymbol{y}} \doteq \boldsymbol{y} - G_{\boldsymbol{\theta}^0}(\boldsymbol{z})$, $\boldsymbol{J} \doteq \boldsymbol{J}_G(\boldsymbol{\theta}^0)$, and $\boldsymbol{c} \doteq \boldsymbol{\theta} - \boldsymbol{\theta}^0$. So, the least-squares objective in Eq. (4) is equivalent to

$$\|\widehat{\boldsymbol{y}} - \boldsymbol{J}\boldsymbol{c}\|_2^2 \tag{10}$$

and the gradient update reads

$$\boldsymbol{c}^t = \boldsymbol{c}^{t-1} - \eta \boldsymbol{J}^\mathsf{T}\left(\boldsymbol{J}\boldsymbol{c}^{k-1} - \widehat{\boldsymbol{y}}\right), \tag{11}$$

where $\boldsymbol{c}^0 = \boldsymbol{0}$ and $\boldsymbol{x}^t = \boldsymbol{J}\boldsymbol{c}^t + G_{\boldsymbol{\theta}^0}(\boldsymbol{z})$. The residual at time $t$ can be computed as

$$\boldsymbol{r}^t \doteq \widehat{\boldsymbol{y}} - \boldsymbol{J}\boldsymbol{c}^t \tag{12}$$

$$= \widehat{\boldsymbol{y}} - \boldsymbol{J}\left(\boldsymbol{c}^{t-1} - \eta\boldsymbol{J}^\mathsf{T}\left(\boldsymbol{J}\boldsymbol{\theta}^{t-1} - \widehat{\boldsymbol{y}}\right)\right) \tag{13}$$

$$= \left(\boldsymbol{I} - \eta\boldsymbol{J}\boldsymbol{J}^\mathsf{T}\right)\left(\widehat{\boldsymbol{y}} - \boldsymbol{J}\boldsymbol{c}^{t-1}\right) \tag{14}$$

$$= \left(\boldsymbol{I} - \eta\boldsymbol{J}\boldsymbol{J}^\mathsf{T}\right)^2\left(\widehat{\boldsymbol{y}} - \boldsymbol{J}\boldsymbol{c}^{t-2}\right) = \dots \tag{15}$$

$$= \left(\boldsymbol{I} - \eta\boldsymbol{J}\boldsymbol{J}^\mathsf{T}\right)^t\left(\widehat{\boldsymbol{y}} - \boldsymbol{J}\boldsymbol{c}^0\right) \quad (\text{using } \boldsymbol{c}^0 = \boldsymbol{0}) \tag{16}$$

$$= \left(\boldsymbol{I} - \eta\boldsymbol{J}\boldsymbol{J}^\mathsf{T}\right)^t\widehat{\boldsymbol{y}}. \tag{17}$$

Assume that the SVD of $\boldsymbol{J}$ is as $\boldsymbol{J} = \boldsymbol{W}\boldsymbol{\Sigma}\boldsymbol{V}^\mathsf{T}$. Then

$$\boldsymbol{r}^t = \left(\boldsymbol{I} - \eta\boldsymbol{W}\boldsymbol{\Sigma}^2\boldsymbol{W}^\mathsf{T}\right)^t\widehat{\boldsymbol{y}} = \sum_i \left(1 - \eta\sigma_i^2\right)^t \boldsymbol{w}_i^\mathsf{T}\widehat{\boldsymbol{y}}\boldsymbol{w}_i \tag{18}$$

and so

$$\boldsymbol{J}\boldsymbol{c}^t = \widehat{\boldsymbol{y}} - \boldsymbol{r}^t = \sum_i \left(1 - \left(1 - \eta\sigma_i^2\right)^t\right) \boldsymbol{w}_i^\mathsf{T}\widehat{\boldsymbol{y}}\boldsymbol{w}_i. \tag{19}$$

Consider a set of $W$ vectors $\mathcal{V} = \{\boldsymbol{v}_1, \dots, \boldsymbol{v}_W\}$. We have the empirical variance.

$$\text{VAR}(\mathcal{V}) = \frac{1}{W}\sum_{w=1}^{W}\left\|\boldsymbol{v}_w - \frac{1}{W}\sum_{j=1}^{W}\boldsymbol{v}_j\right\|_2^2 = \frac{1}{W}\sum_{w=1}^{W}\|\boldsymbol{v}_w\|_2^2 - \left\|\frac{1}{W}\sum_{w=1}^{W}\boldsymbol{v}_w\right\|_2^2. \tag{20}$$

Therefore, the variance of the set $\{\boldsymbol{x}^t, \boldsymbol{x}^{t+1}, \dots, \boldsymbol{x}^{t+W-1}\}$, same as the variance of the set $\{\boldsymbol{J}\boldsymbol{c}^t, \boldsymbol{J}\boldsymbol{c}^{t+1}, \dots, \boldsymbol{J}\boldsymbol{c}^{t+W-1}\}$, can be calculated as

$$\frac{1}{W}\sum_{w=0}^{W-1}\sum_i (\boldsymbol{w}_i^\mathsf{T}\widehat{\boldsymbol{y}})^2\left(1 - \left(1 - \eta\sigma_i^2\right)^{t+w}\right)^2 - \frac{1}{W^2}\sum_i (\boldsymbol{w}_i^\mathsf{T}\widehat{\boldsymbol{y}})^2\left(\sum_{w=0}^{W-1} 1 - \left(1 - \eta\sigma_i^2\right)^{t+w}\right)^2 \tag{21}$$

$$= \frac{1}{W^2}\sum_i (\boldsymbol{w}_i^\mathsf{T}\widehat{\boldsymbol{y}})^2\left[W\sum_{w=0}^{W-1}\left(1 - \left(1 - \eta\sigma_i^2\right)^{t+w}\right)^2 - \left(\sum_{w=0}^{W-1} 1 - \left(1 - \eta\sigma_i^2\right)^{t+w}\right)^2\right] \tag{22}$$

$$= \frac{1}{W^2}\sum_i (\boldsymbol{w}_i^\mathsf{T}\widehat{\boldsymbol{y}})^2\left[\left(W^2 + W\frac{(1 - \eta\sigma_i^2)^{2t}(1 - (1 - \eta\sigma_i^2)^{2W})}{1 - (1 - \eta\sigma_i^2)^2} - 2W\frac{(1 - \eta\sigma_i^2)^t(1 - (1 - \eta\sigma_i^2)^W)}{\eta\sigma_i^2}\right)\right.$$

$$\left. - \left(W^2 - 2W\frac{(1 - \eta\sigma_i^2)^t(1 - (1 - \eta\sigma_i^2)^W)}{\eta\sigma_i^2} + \frac{(1 - \eta\sigma_i^2)^{2t}\left(1 - (1 - \eta\sigma_i^2)^W\right)^2}{\eta^2\sigma_i^4}\right)\right] \tag{23}$$

$$= \frac{1}{W^2}\sum_i \langle\boldsymbol{w}_i, \widehat{\boldsymbol{y}}\rangle^2 \frac{(1 - \eta\sigma_i^2)^{2t}}{\eta\sigma_i^2}\left[W\frac{1 - (1 - \eta\sigma_i^2)^{2W}}{2 - \eta\sigma_i^2} - \frac{(1 - (1 - \eta\sigma_i^2)^W)^2}{\eta\sigma_i^2}\right]. \tag{24}$$

So the constants $C_{W,\eta,\sigma_i}$'s are defined as

$$C_{W,\eta,\sigma_i} \doteq \frac{1}{W^2\eta\sigma_i^2}\left[W\frac{1-(1-\eta\sigma_i^2)^{2W}}{2-\eta\sigma_i^2} - \frac{(1-(1-\eta\sigma_i^2)^W)^2}{\eta\sigma_i^2}\right]. \tag{25}$$

To see they are nonnegative, it is sufficient to show that

$$W\frac{1-(1-\eta\sigma_i^2)^{2W}}{2-\eta\sigma_i^2} - \frac{(1-(1-\eta\sigma_i^2)^W)^2}{\eta\sigma_i^2} \geq 0$$

$$\Longleftrightarrow \eta\sigma_i^2 W\left(1-(1-\eta\sigma_i^2)^{2W}\right) - \left(2-\eta\sigma_i^2\right)(1-(1-\eta\sigma_i^2)^W)^2 \geq 0. \tag{26}$$

Now consider the function.

$$h(\xi, W) = \xi W\left(1-(1-\xi)^{2W}\right) - (2-\xi)(1-(1-\xi)^W)^2 \quad \xi \in [0,1], W \geq 1. \tag{27}$$

First, one can easily check that $\partial_W h(\xi, W) \geq 0$ for all $W \geq 1$ and all $\xi \in [0,1]$, that is, $h(\xi, W)$ increases monotonically with respect to $W$. Thus, to prove $C_{W,\eta,\sigma_i} \geq 0$, it suffices to show that $h(\xi, 1) \geq 0$. Now

$$h(\xi, 1) = \xi\left(1-(1-\xi)^2\right) - (2-\xi)\xi^2 = 0, \tag{28}$$

completing the proof. $\qquad\square$

## A.3 Proof of 2.2

We first re-state Theorem 2 in Heckel & Soltanolkotabi (2020b).

**Theorem A.1** (Heckel & Soltanolkotabi (2020b)). *Let $\boldsymbol{x} \in \mathbb{R}^n$ be a signal in the span of the first $p$ trigonometric basis functions, and consider a noisy observation $\boldsymbol{y} = \boldsymbol{x} + \boldsymbol{n}$, where the noise $\boldsymbol{n} \sim \mathcal{N}\left(\boldsymbol{0}, \xi^2/n \cdot \boldsymbol{I}\right)$. To denoise this signal, we fit a two-layer generator network $G_{\boldsymbol{C}}(\boldsymbol{B}) = \mathrm{ReLU}(\boldsymbol{U}\boldsymbol{B}\boldsymbol{C})\boldsymbol{v}$, where $\boldsymbol{v} = [1,\ldots,1,-1,\ldots,-1]/\sqrt{k}$, and $\boldsymbol{B} \sim_{iid} \mathcal{N}(0,1)$, and $\boldsymbol{U}$ is an upsampling operator that implements circular convolution with a given kernel $\boldsymbol{u}$. Denote $\boldsymbol{\sigma} \doteq \|\boldsymbol{u}\|_2|\boldsymbol{F}g(\boldsymbol{u} \circledast \boldsymbol{u}/\|\boldsymbol{u}\|_2^2)|^{1/2}$ where $g(t) = (1-\cos^{-1}(t)/\pi)t$ and $\circledast$ denote the circular convolution. Fix any $\varepsilon \in (0, \sigma_p/\sigma_1]$, and suppose that $k \geq C_{\boldsymbol{u}}n/\varepsilon^8$, where $C_{\boldsymbol{u}} > 0$ is a constant depending only on $\boldsymbol{u}$. Consider gradient descent with step size $\eta \leq \|\boldsymbol{F}\boldsymbol{u}\|_\infty^{-2}$ ($\boldsymbol{F}\boldsymbol{u}$ is the Fourier transform of $\boldsymbol{u}$ ) starting from $\boldsymbol{C}_0 \sim_{iid} \mathcal{N}\left(0,\omega^2\right)$, entries $\omega \propto \frac{\|\boldsymbol{y}\|_2}{\sqrt{n}}$. Then, for all iterations $t$ obeying $t \leq \frac{100}{\eta\sigma_p^2}$, the reconstruction error obeys*

$$\|G_{\boldsymbol{C}^t}(\boldsymbol{B}) - \boldsymbol{x}\|_2 \leq \left(1-\eta\sigma_p^2\right)^t \|\boldsymbol{x}\|_2 + \sqrt{\sum_{i=1}^n \left((1-\eta\sigma_i^2)^t - 1\right)^2(\boldsymbol{w}_i^\intercal\boldsymbol{n})^2} + \varepsilon\|\boldsymbol{y}\|_2$$

*with probability at least $1 - \exp\left(-k^2\right) - n^{-2}$.*

Note that since $\boldsymbol{B} \sim_{iid} \mathcal{N}(0,1)$ and hence is full-rank with probability one, the original Theorem 1 & 2 of Heckel & Soltanolkotabi (2020b) rename $\boldsymbol{B}\boldsymbol{C}$ to $\boldsymbol{C}'$ and state the result directly on $\boldsymbol{C}'$, that is, assume that the model is $\mathrm{ReLU}(\boldsymbol{U}\boldsymbol{C}')\boldsymbol{v}$. It is easy to see that the original theorems imply the version stated here.

With this, we can obtain our Theorem 2.2, stated in full technical form here:

**Theorem A.2.** *Let $\boldsymbol{x} \in \mathbb{R}^n$ be a signal in the span of the first $p$ trigonometric basis functions, and consider a noisy observation $\boldsymbol{y} = \boldsymbol{x} + \boldsymbol{n}$, where the noise $\boldsymbol{n} \sim \mathcal{N}\left(\boldsymbol{0}, \xi^2/n \cdot \boldsymbol{I}\right)$. To denoise this signal, we fit a two-layer generator network $G_{\boldsymbol{C}}(\boldsymbol{B}) = \mathrm{ReLU}(\boldsymbol{U}\boldsymbol{B}\boldsymbol{C})\boldsymbol{v}$, where $\boldsymbol{v} = [1,\ldots,1,-1,\ldots,-1]/\sqrt{k}$, and $\boldsymbol{B} \sim_{iid} \mathcal{N}(0,1)$, and $\boldsymbol{U}$ is an upsampling operator that implements circular convolution with a given kernel $\boldsymbol{u}$. Denote $\boldsymbol{\sigma} \doteq \|\boldsymbol{u}\|_2|\boldsymbol{F}g(\boldsymbol{u} \circledast \boldsymbol{u}/\|\boldsymbol{u}\|_2^2)|^{1/2}$ where $g(t) = (1-\cos^{-1}(t)/\pi)t$ and $\circledast$ denotes the circular convolution. Fix any $\varepsilon \in (0, \sigma_p/\sigma_1]$, and suppose $k \geq C_{\boldsymbol{u}}n/\varepsilon^8$, where $C_{\boldsymbol{u}} > 0$ is a constant only depending on $\boldsymbol{u}$. Consider gradient descent with step size $\eta \leq \|\boldsymbol{F}\boldsymbol{u}\|_\infty^{-2}$ ($\boldsymbol{F}\boldsymbol{u}$ is the Fourier transform of $\boldsymbol{u}$ ) starting from $\boldsymbol{C}_0 \sim_{iid} \mathcal{N}\left(0,\omega^2\right)$, entries $\omega \propto \frac{\|\boldsymbol{y}\|_2}{\sqrt{n}}$. Then, for all iterates $t$ obeying $t \leq \frac{100}{\eta\sigma_p^2}$, our WMV obeys*

$$\mathrm{WMV} \leq \frac{12}{W}\|\boldsymbol{x}\|_2^2\frac{\left(1-\eta\sigma_p^2\right)^{2t}}{1-(1-\eta\sigma_p^2)^2} + 12\sum_{i=1}^n \left(\left(1-\eta\sigma_i^2\right)^{t+W-1}-1\right)^2(\boldsymbol{w}_i^\intercal\boldsymbol{n})^2 + 12\varepsilon^2\|\boldsymbol{y}\|_2^2 \tag{29}$$

*with probability at least* $1 - \exp\left(-k^2\right) - n^{-2}$.

*Proof.* We make use of the basic inequality: $\|\boldsymbol{a} - \boldsymbol{b}\|_2^2 \le 2\|\boldsymbol{a}\|_2^2 + 2\|\boldsymbol{b}\|_2^2$ for any two vectors $\boldsymbol{a}, \boldsymbol{b}$ of compatible dimension. We have

$$\frac{1}{W} \sum_{w=0}^{W-1} \|G_{\boldsymbol{C}^{t+w}}(\boldsymbol{B}) - \frac{1}{W} \sum_{j=0}^{W-1} G_{\boldsymbol{C}^{t+j}}(\boldsymbol{B})\|_2^2 \tag{30}$$

$$= \frac{1}{W} \sum_{w=0}^{W-1} \|G_{\boldsymbol{C}^{t+w}}(\boldsymbol{B}) - \boldsymbol{x} + \boldsymbol{x} - \frac{1}{W} \sum_{j=0}^{W-1} G_{\boldsymbol{C}^{t+j}}(\boldsymbol{B})\|_2^2 \tag{31}$$

$$\le \left(\frac{2}{W} \sum_{w=0}^{W-1} \|G_{\boldsymbol{C}^{t+w}}(\boldsymbol{B}) - \boldsymbol{x}\|_2^2\right) + 2\|\boldsymbol{x} - \frac{1}{W} \sum_{j=0}^{W-1} G_{\boldsymbol{C}^{t+j}}(\boldsymbol{B})\|_2^2 \tag{32}$$

$$\le \frac{2}{W} \sum_{w=0}^{W-1} \|G_{\boldsymbol{C}^{t+w}}(\boldsymbol{B}) - \boldsymbol{x}\|_2^2 + \frac{2}{W} \sum_{j=0}^{W-1} \|G_{\boldsymbol{C}^{t+j}}(\boldsymbol{B}) - \boldsymbol{x}\|_2^2 \tag{33}$$

$$(\boldsymbol{z} \mapsto \|\boldsymbol{z} - \boldsymbol{x}\|_2^2 \text{ convex and Jensen's inequality})$$

$$= \frac{4}{W} \sum_{w=0}^{W-1} \|G_{\boldsymbol{C}^{t+w}}(\boldsymbol{B}) - \boldsymbol{x}\|_2^2. \tag{34}$$

In view of Theorem A.1,

$$\|G_{\boldsymbol{C}^{t+w}}(\boldsymbol{B}) - \boldsymbol{x}\|_2^2 \le 3\left(1 - \eta\sigma_p^2\right)^{2t+2w} \|\boldsymbol{x}\|_2^2 + 3\sum_{i=1}^{n} \left(\left(1 - \eta\sigma_j^2\right)^{t+w} - 1\right)^2 (\boldsymbol{w}_i^{\mathsf{T}}\boldsymbol{n})^2 + 3\varepsilon^2\|\boldsymbol{y}\|_2^2. \tag{35}$$

Thus,

$$\sum_{w=0}^{W-1} \|G_{\boldsymbol{C}^{t+w}}(\boldsymbol{B}) - \boldsymbol{x}\|_2^2$$

$$\le 3\|\boldsymbol{x}\|_2^2 \sum_{w=0}^{W-1} \left(1 - \eta\sigma_p^2\right)^{2t+2w} + 3 \sum_{w=0}^{W-1} \sum_{i=1}^{n} \left(\left(1 - \eta\sigma_i^2\right)^{t+w} - 1\right)^2 (\boldsymbol{w}_i^{\mathsf{T}}\boldsymbol{n})^2 + 3W\varepsilon^2\|\boldsymbol{y}\|_2^2 \tag{36}$$

$$\le 3\|\boldsymbol{x}\|_2^2 \frac{\left(1 - \eta\sigma_p^2\right)^{2t}\left(1 - (1 - \eta\sigma_p^2)^{2W}\right)}{1 - (1 - \eta\sigma_p^2)^2} + 3W \sum_{i=1}^{n} \left(\left(1 - \eta\sigma_i^2\right)^{t+W-1} - 1\right)^2 (\boldsymbol{w}_i^{\mathsf{T}}\boldsymbol{n})^2 + 3W\varepsilon^2\|\boldsymbol{y}\|_2^2 \tag{37}$$

$$\le 3\|\boldsymbol{x}\|_2^2 \frac{\left(1 - \eta\sigma_p^2\right)^{2t}}{1 - (1 - \eta\sigma_p^2)^2} + 3W \sum_{i=1}^{n} \left(\left(1 - \eta\sigma_i^2\right)^{t+W-1} - 1\right)^2 (\boldsymbol{w}_i^{\mathsf{T}}\boldsymbol{n})^2 + 3W\varepsilon^2\|\boldsymbol{y}\|_2^2, \tag{38}$$

completing the proof. $\qquad\square$

### A.4   ES-EMV algorithm

The exponential moving variance version of our method is summarized in Algorithm 2.

### A.5   More details on major DIP variants

**Deep Decoder (DD)**   (Heckel & Hand, 2019) differs from DIP mainly in terms of network architecture: It is typically a *under-parameterized* network consisting mainly of $1 \times 1$ convolutions, upsampling, ReLU and channel-wise normalization layers, while DIP uses an *over-parameterized*, U-net like convolutional network.

---

**Algorithm 2** DIP with ES–EMV

---

**Input:** random seed $\boldsymbol{z}$, randomly-initialized $G_{\boldsymbol{\theta}}$, forgetting factor $\alpha \in (0,1)$, patience number $P$, iteration
counter $k = 0$, $\text{EMA}^0 = 0$, $\text{EMV}^0 = 0$, $\text{EMV}_{\min} = \infty$

**Output:** reconstruction $\boldsymbol{x}^*$

 1: **while** not stopped **do**
 2:     update $\boldsymbol{\theta}$ via Eq. (2) to obtain $\boldsymbol{\theta}^{k+1}$ and $\boldsymbol{x}^{k+1}$
 3:     $\text{EMA}^{k+1} = (1 - \alpha)\text{EMA}^k + \alpha \boldsymbol{x}^{k+1}$
 4:     $\text{EMV}^{k+1} = (1 - \alpha)\text{EMV}^k + \alpha(1 - \alpha)\|\boldsymbol{x}^{k+1} - \text{EMA}^k\|_2^2$
 5:     **if** $\text{EMV}^{k+1} < \text{EMV}_{\min}$ **then**
 6:         $\text{EMV}_{\min} \leftarrow \text{EMV}^{k+1}$, $\boldsymbol{x}^* \leftarrow \boldsymbol{x}^{k+1}$
 7:     **end if**
 8:     **if** $\text{EMV}_{\min}$ stagnates for $P$ iterations **then**
 9:         stop and return $\boldsymbol{x}^*$
10:     **end if**
11:     $k = k + 1$
12: **end while**

---

**GP-DIP**   (Cheng et al., 2019) uses the original DIP (Ulyanov et al., 2018) network and formulation, but replaces stochastic gradient descent (SGD) by stochastic gradient Langevin dynamics (SGLD) in the gradient update step. i.e., for the generic gradient step for optimizing Eq. (2) reads:

$$\boldsymbol{\theta}^+ = \boldsymbol{\theta} - t\nabla_{\boldsymbol{\theta}}[\ell(\boldsymbol{y}, f(G_{\boldsymbol{\theta}}(\boldsymbol{z}))) + \lambda R(G_{\boldsymbol{\theta}}(\boldsymbol{z}))] + \eta \tag{39}$$

where $\eta$ is zero-mean Gaussian with an isotropic variance level $t$.

**DIP-TV**   (Cascarano et al., 2021) uses the original DIP (Ulyanov et al., 2018) network, with a Total Variation (TV) regularizer added. Then, the proposed objective is solved with the Alternating Direction Method of Multipliers (ADMM) framework.

**SIREN**   (Sitzmann et al., 2020) treats the object directly as a continuous function on $\mathbb{R}^2$ or $\mathbb{R}^3$ (or higher-dimensional spaces depending on the application) and hence parameterizes it as a multi-layer perceptron (MLP): 1) the input to SIREN is the 2D/3D coordinate of each pixel instead of random values, and 2) the network uses a sinusoidal activation function instead of the commonly used ReLU. When substituting the DIP network with SIREN and solve Eq. (2) problems, similar overfitting issue is still observed.

### A.6   More details on major ES methods

Here, we provide more details on the main competing methods.

**Spectral Bias (SB)**   Shi et al. (2022) operates on DD models and proposes two modifications to change the spectral bias: (1) controlling the operator norm of the weight $\boldsymbol{w}$ for each convolutional layer by normalization

$$\boldsymbol{w}' = \frac{\boldsymbol{w}}{\max\left(1, \|\boldsymbol{w}\|_{\text{op}}/\lambda\right)}, \tag{40}$$

ensuring that $\|\boldsymbol{w}'\|_{\text{op}} \leq \lambda$, which in turn controls the Fourier spectrum of the underlying function represented by the layer; (2) performing Gaussian upsampling instead of the typical bilinear upsampling to suppress the smoothness effect of the latter. These two modifications with appropriate parameter setting ($\lambda$, and $\sigma$ in Gaussian filtering) can improve the learning of the high-frequency components by DD, and allow the blurriness-over-sharpness stopping criterion.

$$\Delta r(\boldsymbol{x}^t) = \frac{1}{W}\left|\sum_{w=1}^{W} r(\boldsymbol{x}^{t-w}) - \sum_{w=1}^{W} r(\boldsymbol{x}^{t-W-w})\right|, \tag{41}$$

where $r(\boldsymbol{x}') = B(\boldsymbol{x}')/S(\boldsymbol{x}')$, and $B(\cdot)$ and $S(\cdot)$ are the blurriness and sharpness metrics in Crete et al. (2007) and Bahrami & Kot (2014), respectively. In other words, the criterion in Eq. (41) measures the change in the average blurriness-over-sharpness ratios in consecutive windows of size $W$, and small changes indicate good ES points. But, as mentioned, this criterion only works for modified DD models and not for other DIP variants, as acknowledged by the authors in Shi et al. (2022) and confirmed in our experiment (see Sec. 3.1).

**DF-STE**    Jo et al. (2021) targets Gaussian denoising with known noise levels (i.e. $\boldsymbol{y} = \boldsymbol{x} + \boldsymbol{n}$, where $n$ is the i.i.d. Gaussian noise) and considers the objective.

$$\min_{\boldsymbol{\theta}} \frac{1}{n^2}\|\boldsymbol{y} - G_{\boldsymbol{\theta}}(\boldsymbol{y})\|_F^2 + \frac{\sigma^2}{n^2} \operatorname{tr} \boldsymbol{J}_{G_{\boldsymbol{\theta}}}(\boldsymbol{y}), \tag{42}$$

where $\operatorname{tr} \boldsymbol{J}_{G_{\boldsymbol{\theta}}}(\boldsymbol{y})$ is the trace of the network Jacobian with respect to the input, that is, the divergence term in Jo et al. (2021). The divergence term is a proxy for controlling the capacity of the network. The paper then proposes a heuristic zero-crossing stopping criterion that stops the iteration when the loss starts to cross zero into negative values. Although the idea works reasonably well on Gaussian denoising with low and known noise level (the variance level $\sigma^2$ is explicitly needed in the regularization parameter ahead of the divergence term), it starts to break down when the noise level increases even if the right noise level is provided; see Sec. 3.1. Also, although the paper has extended the formulation to handle Poisson noise, it is unclear how to generalize the idea for handling other types of noise, as well as how to move beyond simple additive denoising problems.

**SV-ES**    Li et al. (2021) proposes training an autoencoder online using the reconstruction sequence $\{\boldsymbol{x}^t\}_{t\geq 1}$:

$$\min_{\boldsymbol{w}, \boldsymbol{v}} \sum_{t\geq 1} \ell_{\mathrm{AE}}\big(\boldsymbol{x}^t, D_{\boldsymbol{w}} \circ E_{\boldsymbol{v}}(\boldsymbol{x}^t)\big). \tag{43}$$

Any new $\boldsymbol{x}^t$ passes through the current autoencoder and the reconstruction error $\ell_{\mathrm{AE}}$ is recorded. They observe that the error curve typically follows a U-shaped shape and that the valley of the curve is approximately aligned with the peak of the PNSR curve. Therefore, they design an ES method by detecting the valley of the error curve. This method works reasonably well for different IPs and different DIP variants. A major drawback is efficiency: the overhead caused by the online training of the autoencoder is on an order of magnitude higher than the cost of the DIP update itself, as shown in Tab. 3.

**DOP**    You et al. (2020) considers only additive sparse noise (e.g., salt and pepper noise) and proposes modeling the clean image and noise explicitly in the objective:

$$\min_{\boldsymbol{\theta}, \boldsymbol{g}, \boldsymbol{h}} \|\boldsymbol{y} - G_{\boldsymbol{\theta}}(\boldsymbol{z}) - (\boldsymbol{g} \circ \boldsymbol{g} - \boldsymbol{h} \circ \boldsymbol{h})\|_F^2, \tag{44}$$

where the overparameterized term $\boldsymbol{g} \circ \boldsymbol{g} - \boldsymbol{h} \circ \boldsymbol{h}$ ($\circ$ denotes the Hadamard product) is meant to capture sparse noise, where a similar idea has been shown to be effective for sparse recovery in Vaskevicius et al. (2019). Different properly tuned learning rates for the clean image and sparse noise terms are necessary for success. The downside includes the prolonged running time, as it pushes the peak reconstruction to the very last iteration, and the difficulty to extend the idea to other types of noise.

### A.7   Additional experimental details & results

### A.7.1   External codes

- DIP: `https://github.com/DmitryUlyanov/deep-image-prior`
- DD: `https://github.com/reinhardh/supplement_deep_decoder`
- DIP-TV: `https://github.com/sedaboni/ADMM-DIPTV`
- GP-DIP: `https://people.cs.umass.edu/~zezhoucheng/gp-dip/`
- DF-STE: `https://github.com/gistvision/dip-denosing`

- SV-ES: `https://github.com/sun-umn/Self-Validation`
- DOP: `https://github.com/ChongYou/robust-image-recovery`
- SB: `https://github.com/shizenglin/Measure-and-Control-Spectral-Bias`
- CBSD68: `https://github.com/clausmichele/CBSD68-dataset`

### A.7.2 Experiment Settings

Our default setup for all experiments is as follows. Our DIP model is the original from Ulyanov et al. (2018); the optimizer is ADAM with a learning rate 0.01. For all other models, we use their default architectures, optimizers, and hyperparameters. For ES-WMV, the default window size $W = 100$, and the patience number $P = 1000$. We use both PSNR and SSIM to access the reconstruction quality and report PSNR and SSIM gaps (the difference between our detected and peak numbers) as an indicator of our detection performance. **For most experiments, we repeat the experiments 3 times to report the mean and standard deviation**; when not, we explain why.

**Noise generation** Following the noise generation rules of Hendrycks & Dietterich (2019)[2], we simulate four types of noise and three intensity levels for each type of noise. The detailed information is as follows.

- **Gaussian noise:** 0 mean additive Gaussian noise with variance 0.12, 0.18 and 0.26 for low, medium and high noise levels, respectively;

- **Impulse noise:** also known as salt-and-pepper noise, replacing each pixel with probability $p \in [0, 1]$ in a white or black pixel with half chance each. Low, medium and high noise levels correspond to $p = 0.3, 0.5, 0.7$, respectively;

- **Speckle noise:** for each pixel $x \in [0, 1]$, the noisy pixel is $x(1 + \varepsilon)$, where $\varepsilon$ is zero-mean Gaussian with a variance level 0.20, 0.35, 0.45 for low, medium, and high noise levels, respectively;

- **Shot noise:** also known as Poisson noise. For each pixel, $x \in [0, 1]$, the noisy pixel is Poisson distributed with the rate $\lambda x$, where $\lambda$ is $25, 12, 5$ for low, medium, and high noise levels, respectively.

### A.7.3 Denoising examples

We explore the possibility of using the fitting loss for ES here, but we are unable to find correlations between the trend of the loss and that of the PSNR curve, shown in Fig. 18

### A.7.4 Comparison with baseline methods

To further compare with baseline methods, we report the PSNR gaps in high-level noise cases and the SSIM gaps in low- and high-level noise cases in Fig. 19,Fig. 20 and Fig. 21, respectively, which show a trend similar to the results of PSNR gaps. The detection gaps of our method are very marginal ($< 0.02$) for most types and levels of noise (except Baboon and Kodak1 for certain types / levels of noise), while the baseline methods can easily exceed 0.1 for most cases. In addition, we provide some visual detection results in Figs. 7 and 8. Our ES-WMV significantly outperforms the four baseline methods visually.

### A.7.5 Comparison with competing methods

Comparison between ES-WMV with DF-STE for Gaussian and shot noise on the 9 image dataset in terms of SSIM is reported in Fig. 22. Furthermore, we also test our ES-WMV and DF-STE on CBSD68 in Tab. 7. Our ES-WMV wins in high-level noise cases but lags behind DF-STE in the low-level cases. The gaps between our ES-WMV and DF-STE for all noise levels mostly come from the peak performance between the original DIP and DF-STE—modifications in DF-STE have affected peak performance, positively for low-level cases and negatively for high-level cases, not much from our ES method, as evident from the uniformly small detection

---

[2]`https://github.com/hendrycks/robustness`

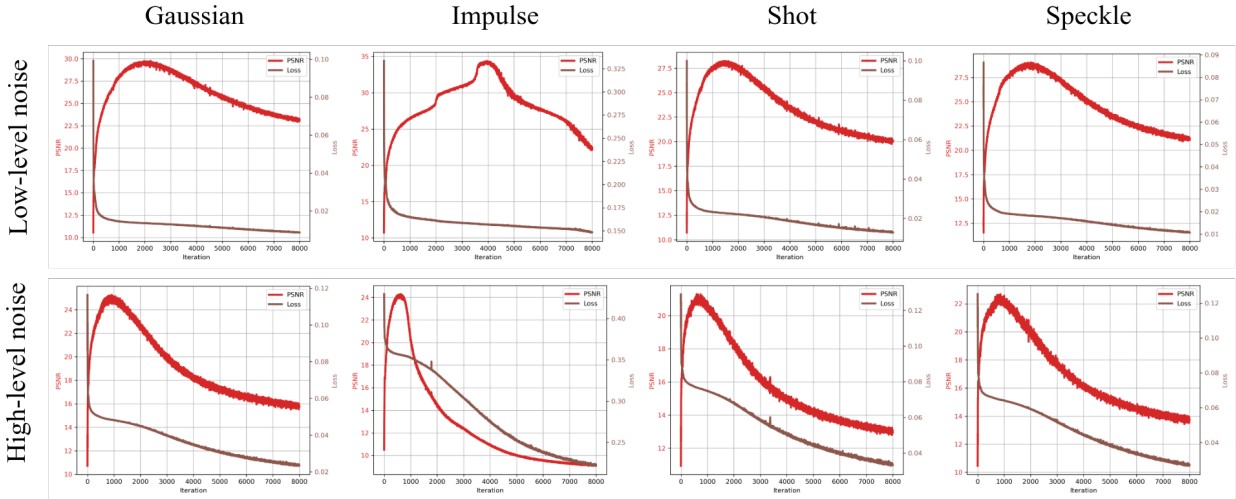

Figure 18: Our ES-WMV method on DIP for denoising "F16" with different noise types and levels (top: low-level noise; bottom: high-level noise). Red curves are PSNR curves, and brown curves are loss curves.

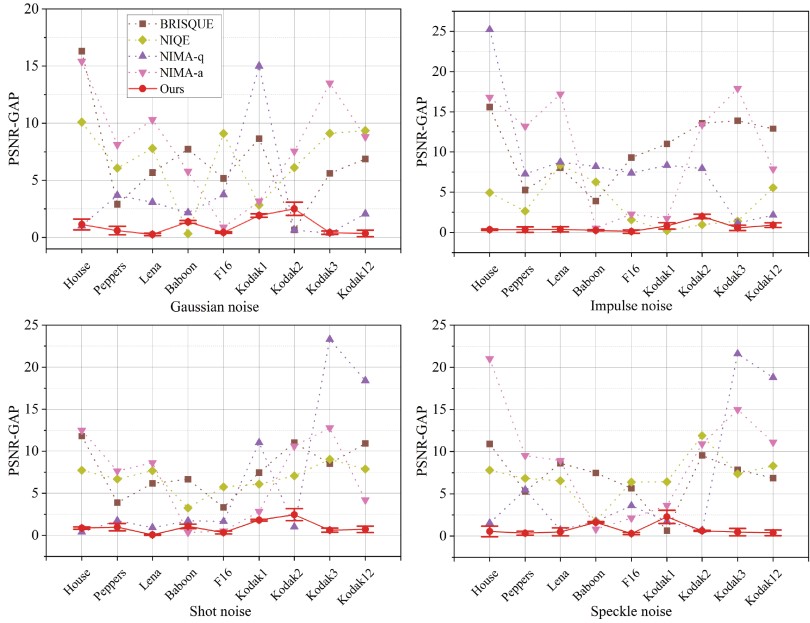

Figure 19: **High-level noise** detection performance in terms of PSNR gaps. For NIMA, we report both technical quality assessment (NIMA-q) and aesthetic assessment (NIMA-a). Smaller PSNR gaps are better.

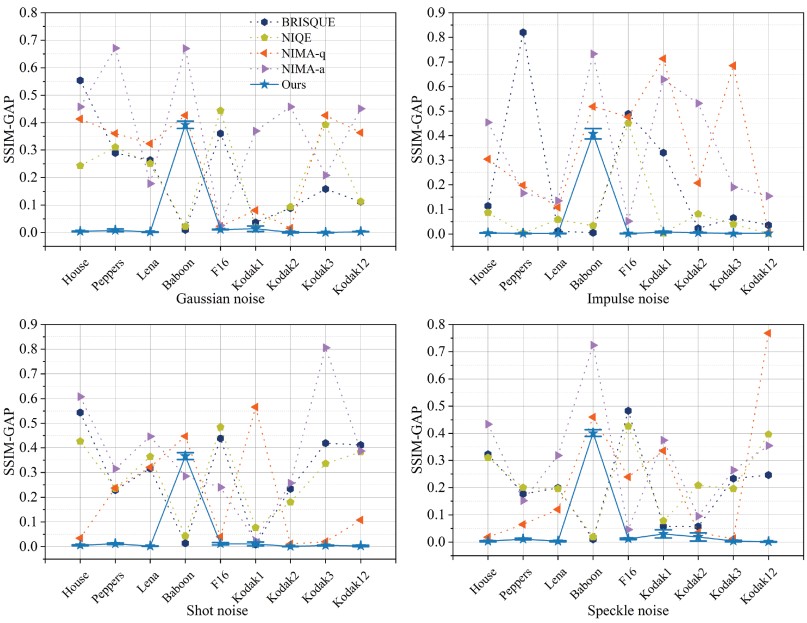

Figure 20: **Low-level noise** detection performance in terms of SSIM gaps. For NIMA, we report both technical quality assessment (NIMA-q) and aesthetic assessment (NIMA-a). Smaller SSIM gaps are better.

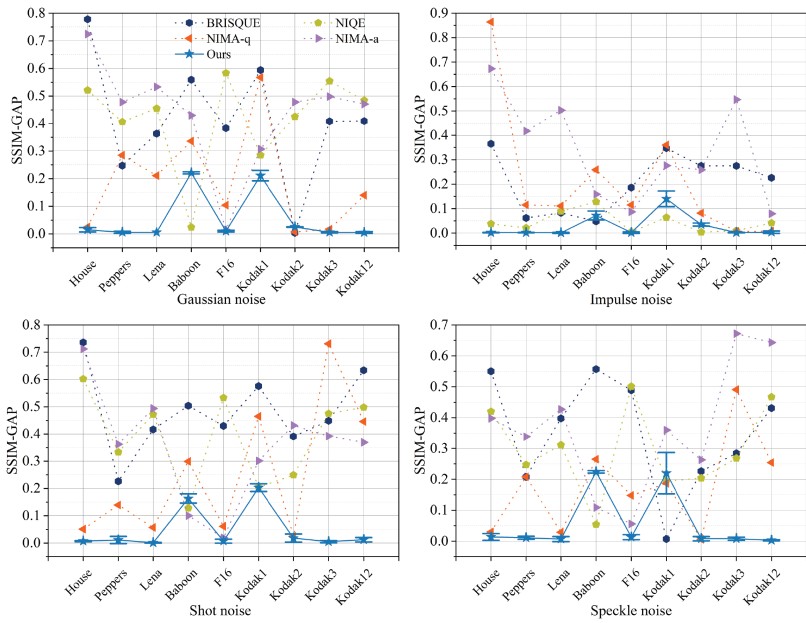

Figure 21: **High-level noise** detection performance in terms of SSIM gaps. For NIMA, we report both technical quality assessment (NIMA-q) and aesthetic assessment (NIMA-a). Smaller SSIM gaps are better.

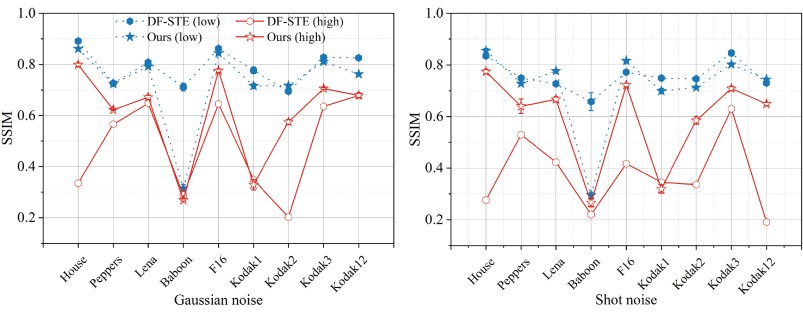

Figure 22: Comparison of DF-STE and ES-WMV for Gaussian and shot noise in terms of SSIM.

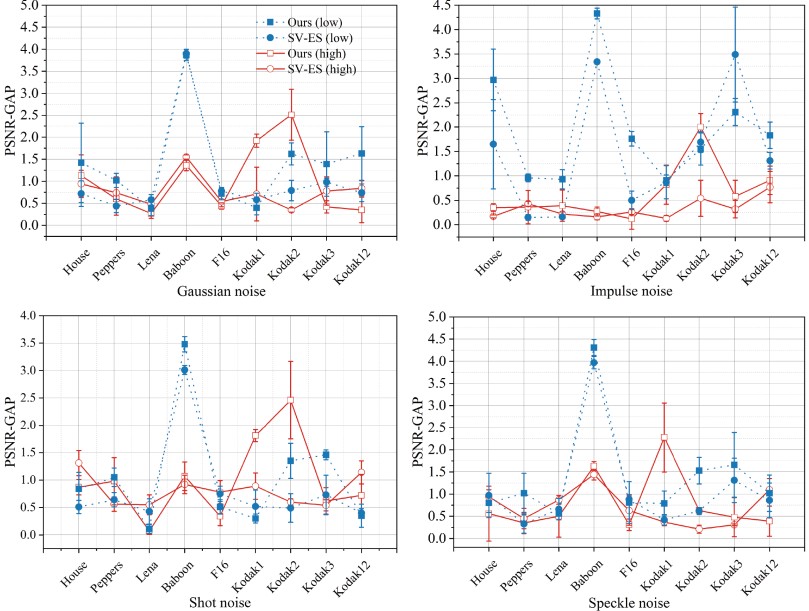

Figure 23: **Low- and high-level noise** detection performance of SV-ES and ours in terms of PSNR gaps.

gaps reported in Tab. 7. Moreover, DF-STE can only handle Gaussian and Poisson noise for denoising, and the exact noise level is a required hyperparameter for their method to work.

Then we compare our ES-WMV and SV-ES in Fig. 23. The DIP results with ES-WMV versus DOP in impulse noise are shown in Tab. 8. For SB, part of the qualitative detection results on the 9 images[3] are reported in Fig. 24.

Table 7: Comparison between ES-WMV and DF-STE for image denoising on the CBSD68 dataset with varying noise level $\sigma$: mean and (std). PSNR gaps below 1.0 are colored as red.

|  | $\sigma = 15$ | $\sigma = 25$ | $\sigma = 50$ |
|---|---|---|---|
| ES-WMV | 28.7(3.2) | 27.4(2.6) | 24.2(2.3) |
| DIP (Peak) | 29.7(3.0) | 28.0(2.4) | 24.9(2.3) |
| PSNR Gap | 1.0(0.7) | 0.7(0.5) | 0.7(0.5) |
| DF-STE | 31.4(1.8) | 28.4(2.2) | 21.1(2.5) |

---

[3]http://www.cs.tut.fi/~foi/GCF-BM3D/index.html#ref_results

Table 8: DIP with ES-WMV vs. DOP on impulse noise: mean and (std).

|  | Low Level | | High Level | |
|---|---|---|---|---|
|  | PSNR | SSIM | PSNR | SSIM |
| DIP-ES | 31.64 (5.69) | 0.85 (0.18) | 24.74 (3.23) | 0.67 (0.19) |
| DOP | 32.12 (4.52) | 0.92 (0.07) | 27.34 (3.78) | 0.86 (0.10) |

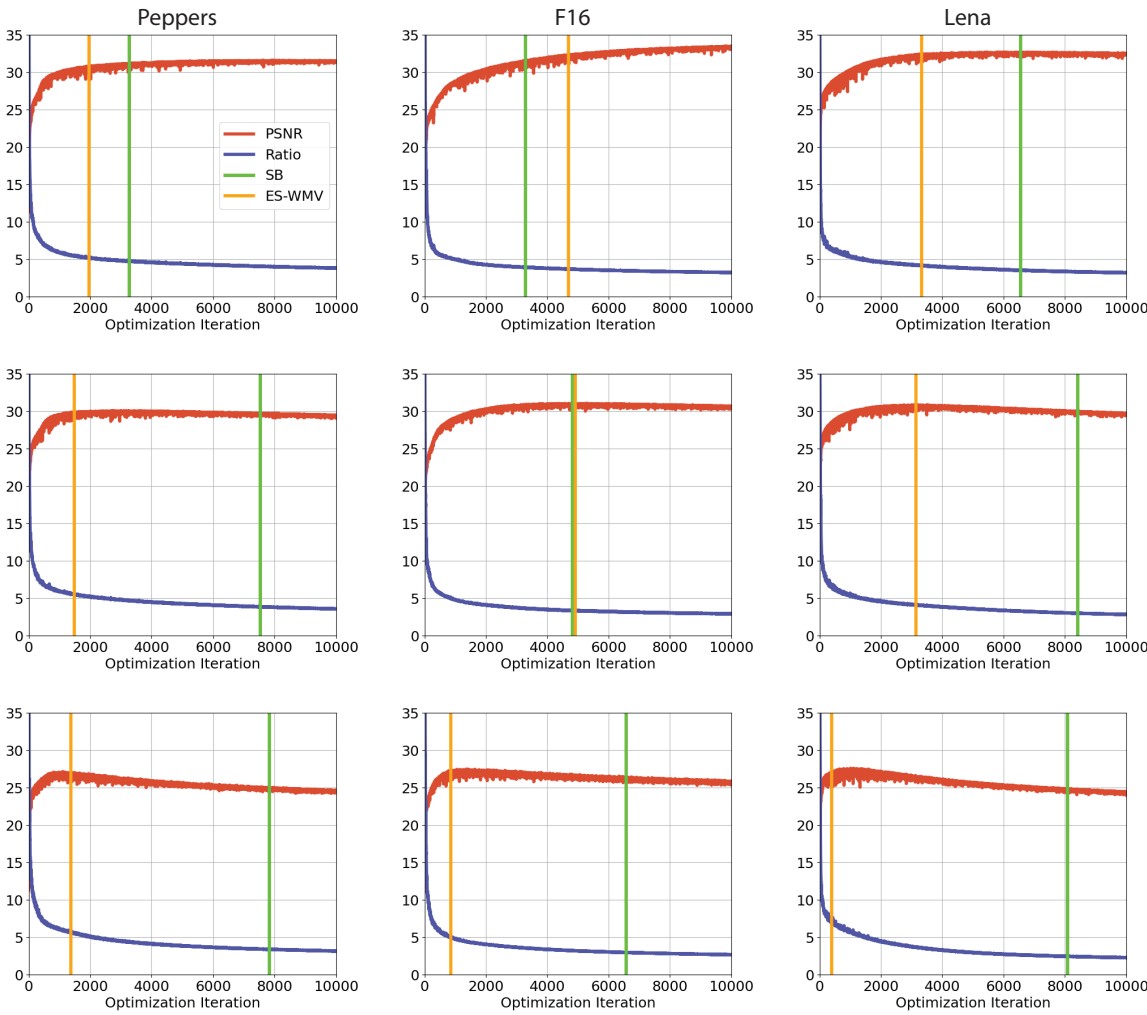

Figure 24: Comparison between ES-WMV and SB for image denoising (top: $\sigma = 15$; middle: $\sigma = 25$; bottom: $\sigma = 50$). The red and blue curves are the PNSR and the ratio metric curves. The orange and green bars indicate the ES points detected by our ES-WMV and SB, respectively.

### A.7.6 ES-WMV as a helper

Performance of ES-WMV on DD, GP-DIP, DIP-TV, and SIREN for Gaussian denoising in terms of SSIM gaps (see Fig. 25).

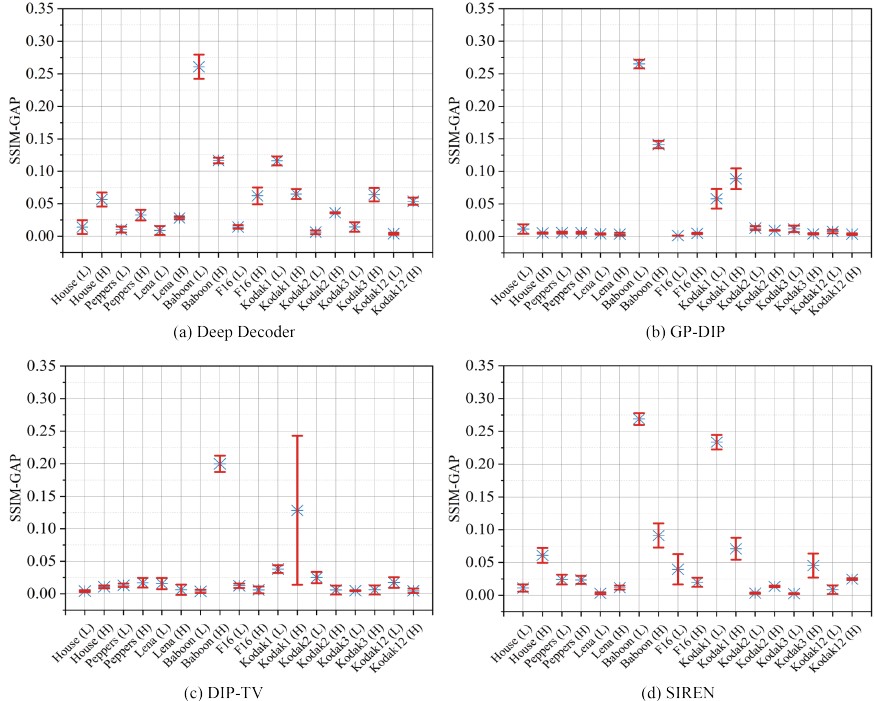

Figure 25: Performance of ES-WMV on DD, GP-DIP, DIP-TV, and SIREN for Gaussian denoising in terms of SSIM gaps. L: low noise level; H: high noise level

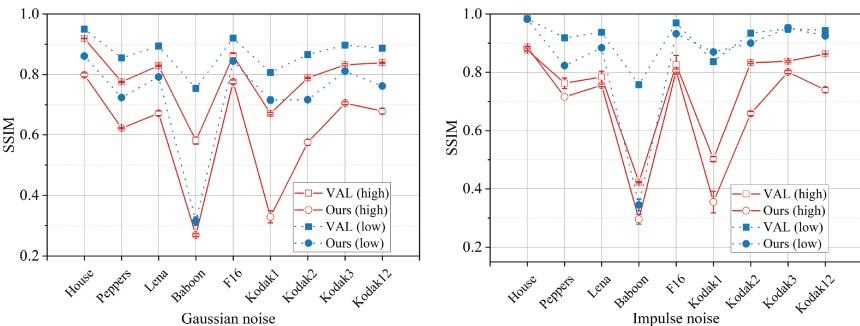

Figure 26: Comparison of VAL and ES-WMV for Gaussian and impulse noise in terms of SSIM.

### A.7.7 Performance on real-world denoising

We randomly sample 1024 images from the RGB track of the NTIRE 2020 Real Image Denoising Challenge (Abdelhamed et al., 2020), and perform DIP-based image denoising. Histograms of PSNR and SSIM gaps are shown in Fig. 27. For DIP with the three different losses, there are only 4.79%, 4.69% and 4.40% images, respectively, whose PSNR gaps are larger than $2dB$.

As stated from the beginning, ES-WMV is designed with real-world IPs, targeting unknown noise types and levels. Given the encouraging performance above, we test it on a common real-world denoising dataset—

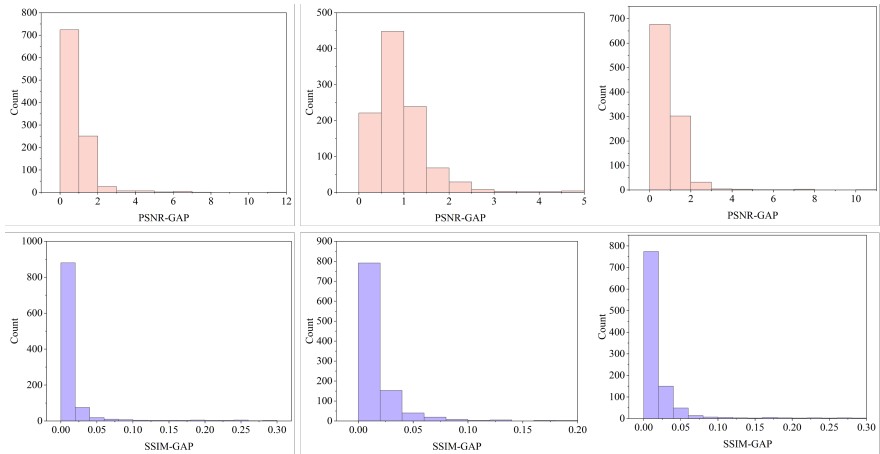

Figure 27: Image denoising of DIP + ES-WMV on the RGB track of the NTIRE 2020 Real Image Denoising Challenge. Top row: histograms of PSNR gaps for DIP (MSE), DIP ($\ell_1$) and DIP (Huber), respectively; bottom row: histograms of SSIM gaps for DIP (MSE), DIP ($\ell_1$) and DIP (Huber), respectively.

Table 9: DIP with ES-WMV on real image denoising on the PolyU Dataset: mean and (std). (**D**: Detected)

|  | PSNR(**D**) | PSNR Gap | SSIM(**D**) | SSIM Gap |
|---|---|---|---|---|
| DIP (MSE) | 36.83 (3.07) | 1.26 (1.22) | 0.98 (0.02) | 0.01 (0.01) |
| DIP ($\ell_1$) | 36.20 (2.81) | 1.64 (1.58) | 0.97 (0.02) | 0.01 (0.01) |
| DIP (Huber) | 36.76 (2.96) | 1.28 (1.09) | 0.98 (0.02) | 0.01 (0.01) |

PolyU Dataset Xu et al. (2018), which contains 100 cropped regions of $512 \times 512$ from 40 scenes. The results are reported in Tab. 9. We do not repeat the experiments here; the means and standard deviations are obtained over the 100 images of the PolyU dataset. On average, our detection gaps are $\leq 1.64$ in PSNR and $\leq 0.01$ in SSIM for this dataset across various losses. The absolute PNSR and SSIM detected are surprisingly high.

### A.7.8 Image Inpainting

Table 10: Detection performance of DIP with ES-WMV for image inpainting: mean and (std). PSNR gaps below 1.00 are colored as red; SSIM gaps below 0.05 are colored as blue. (**D**: Detected)

|  | PSNR(**D**) | PSNR Gap | SSIM(**D**) | SSIM Gap |
|---|---|---|---|---|
| Barbara | 21.59 (0.03) | 0.20 (0.03) | 0.67 (0.00) | 0.00 (0.00) |
| Boat | 21.91 (0.10) | 1.16 (0.18) | 0.68 (0.00) | 0.03 (0.01) |
| House | 27.95 (0.33) | 0.48 (0.10) | 0.89 (0.01) | 0.01 (0.00) |
| Lena | 24.71 (0.30) | 0.37 (0.18) | 0.80 (0.00) | 0.01 (0.00) |
| Peppers | 25.86 (0.22) | 0.23 (0.05) | 0.84 (0.01) | 0.02 (0.00) |
| C.man | 25.26 (0.09) | 0.23 (0.14) | 0.82 (0.00) | 0.01 (0.00) |
| Couple | 21.40 (0.44) | 1.21 (0.53) | 0.63 (0.01) | 0.04 (0.02) |
| Finger | 20.87 (0.04) | 0.24 (0.17) | 0.77 (0.00) | 0.01 (0.01) |
| Hill | 23.54 (0.08) | 0.25 (0.11) | 0.70 (0.00) | 0.00 (0.00) |
| Man | 22.92 (0.25) | 0.46 (0.11) | 0.70 (0.01) | 0.01 (0.00) |
| Montage | 26.16 (0.33) | 0.38 (0.26) | 0.86 (0.01) | 0.03 (0.01) |

### A.7.9 ES-WMV vs. ES-EMV

We now consider our memory-efficient version (ES-EMV) as described in Algorithm 2, and compare it with ES-WMV, as shown in Fig. 28. Besides the memory benefit, ES-EMV runs around 100 times faster than ES-WMV, as reported in Tab. 3 and does seem to provide a consistent improvement on the detected PSNRs for image denoising tasks on NTIRE 2020 Real Image Denoising Challenge (Abdelhamed et al., 2020), PolyU dataset Xu et al. (2018) and the classic 9-image dataset (Dabov et al., 2008) (see Tabs. 11 and 12 and Fig. 28), due to the strong smoothing effect (we set $\alpha = 0.1$). In this paper, we prefer to keep it simple and leave systematic evaluations of these variants for future work.

Table 11: Detection performance comparison between DIP with ES-WMV and DIP with ES-EMV for real image denoising on 1024 images from the RGB track of NTIRE 2020 Real Image Denoising Challenge (Abdelhamed et al., 2020): mean and (std). Higher PSNR and SSIM are in red. (**D**: Detected)

|  | PSNR(**D**)-WMV | PSNR(**D**)-EMV | SSIM(**D**)-WMV | SSIM(**D**)-EMV |
|---|---|---|---|---|
| DIP (MSE) | 34.04 (3.68) | 34.96 (3.80) | 0.92 (0.07) | 0.93 (0.07) |
| DIP ($\ell_1$) | 33.92 (4.34) | 34.83 (4.35) | 0.93 (0.05) | 0.94 (0.05) |
| DIP (Huber) | 33.72 (3.86) | 34.72 (4.04) | 0.92 (0.06) | 0.93 (0.06) |

Table 12: Detection performance comparison between DIP with ES-WMV and DIP with ES-EMV for real image denoising on the PolyU dataset Xu et al. (2018): mean and (std). Higher PSNR and SSIM are in red. (**D**: Detected)

|  | PSNR(**D**)-WMV | PSNR(**D**)-EMV | SSIM(**D**)-WMV | SSIM(**D**)-EMV |
|---|---|---|---|---|
| DIP (MSE) | 36.83 (3.07) | 37.32 (3.82) | 0.98 (0.02) | 0.98 (0.03) |
| DIP ($\ell_1$) | 36.20 (2.81) | 36.43 (3.22) | 0.97 (0.02) | 0.97 (0.02) |
| DIP (Huber) | 36.76 (2.96) | 37.21 (3.19) | 0.98 (0.02) | 0.98 (0.02) |

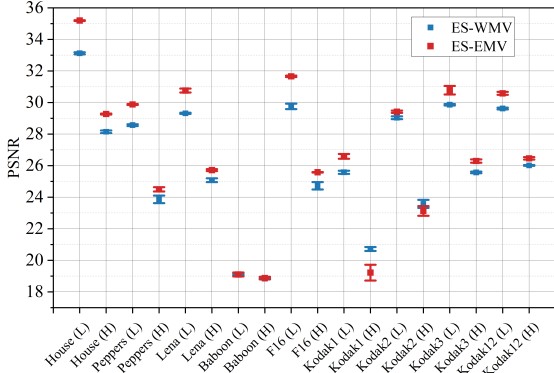

Figure 28: Detected PSNR comparison between DIP with ES-WMV and DIP with ES-EMV on the classic 9-image dataset (Dabov et al., 2008).

### A.7.10 MRI reconstruction

We visualize the performance on two random cases (C1: 1001339 and C2: 1000190 sampled from Darestani & Heckel (2021), part of the fastMRI datatset (Zbontar et al., 2018)) in Fig. 29 (quality measured in SSIM, consistent with Darestani & Heckel (2021)).

### A.7.11 Analysis of failure cases

Our ES-WMV can fail for images with substantial high-frequency components, e.g. Fig. 30.

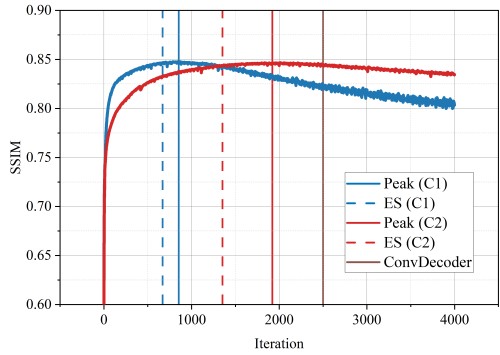

Figure 29: Detection on MRI reconstruction

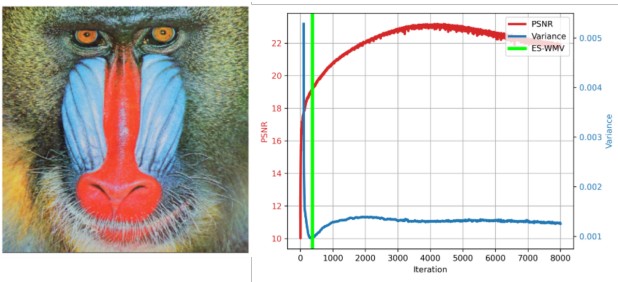

Figure 30: Our ES-WMV method on DIP for denoising "Baboon" with low-level Gaussian noise. Left: clean "Baboon"; right: the denoising process.

### A.7.12 Blind image deblurring (BID)

In this section, we systematically test our ES-WMV and VAL on the entire standard Levin dataset for both low-level and high-level cases. We set the maximum number of iterations to $10,000$ to ensure sufficient optimization. The detected images of our ES-WMV are substantially better than those of VAL, as shown in Tab. 13.

Table 13: BID detection comparison between ES-WMV and VAL on the Levin dataset for both low-level and high-level noise: mean and (std).Higher PSNR is in red and higher SSIM is in blue. (**D**: Detected)

|     | Low Level | | High Level | |
| --- | --- | --- | --- | --- |
|     | PSNR(**D**) | SSIM(**D**) | PSNR(**D**) | SSIM(**D**) |
| WMV | 28.54(0.61) | 0.83(0.04) | 26.41(0.67) | 0.76(0.04) |
| VAL | 18.87(1.44) | 0.50(0.09) | 16.69(1.39) | 0.44(0.10) |

### A.7.13 Ablation study

We vary the window size $W$ (default 100) and patience number $P$ (default: 1000) across a range and check how the detection gap changes for Gaussian denoising with medium-level noise on the classic 9-image dataset (see:Fig. 31).

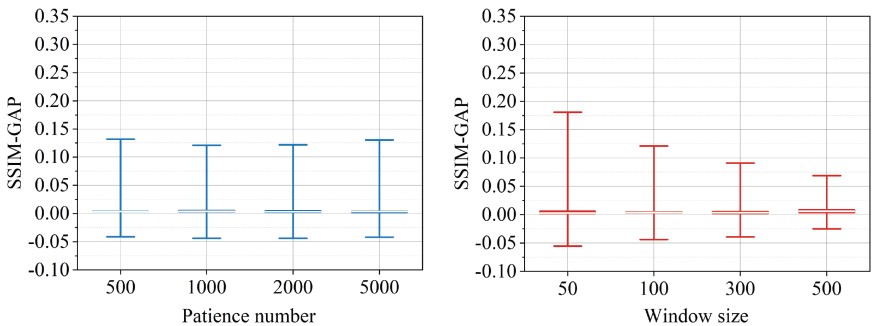

Figure 31: Effect of patience number and window size on detection in terms of SSIM gaps

