# OpenReview forum: "Early Stopping for Deep Image Prior"
_TMLR — Rejected by TMLR_

### Review · Reviewer_XgzK · 2023-02-27

**Summary Of Contributions:**

This paper mainly focuses on the overfitting issue in some deep image prior (DIP) models. To find the near-peak performance without the ground truth, the authors propose an early stopping (ES) strategy that detects near-peak performance across several CI tasks and DIP variants. Experiments demonstrate that based on the running variance of the DIP intermediate reconstructions, the proposed ES strategy is able to outperform other methods.

**Audience:**

No

**Claims And Evidence:**

Yes

**Requested Changes:**

Please refer to the weaknesses above. The authors should highlight the significance of the research direction, carefully state the relationship between the proposed ES strategy and the theorems, and show more results to make the experiments to be convincing.

**Strengths And Weaknesses:**

Strengths:
The proposed ES strategy is simple to calculate the running variance of the reconstruction sequence such that the running trend can be monitored.  Besides, the strategy is robust to the hyperparameters which can be fixed in the experiments.

Weaknesses:
1. The significance of the research direction can be highlighted. The overfitting issue seems to occur in specific kinds of DIP models. Recent diffusion models have outstanding performance on zero-shot image restoration tasks without the overfitting issue. With the fixed interactions, the diffusion model is able to denoise images iteratively. Thus, for the diffusion models, the ES strategy may be not necessary.

2. The relationship between the proposed ES strategy and the theorems is not clear. I do not check the correctness of the theorems. It would be better to provide some intuitive understanding of these theorems. Is Theorem 2.1 suitable for most models, e.g., diffusion models? The authors can put the important theorems in the main paper, and leave other theorems in the supplementary materials.

3. The authors show extensive examples of how to detect the ES points. It is also important to provide more visual comparisons in the experiments.

4. It would be better to compare with some SOTA methods. Besides, the authors can move the super-resolution results in the main paper.

---

> ### Author Response · Authors · 2023-03-30
> **Response to Reviewer XgzK (1/2)**
>
> We thank reviewer XgzK for the detailed review and insightful comments.
>
> ### Q1. The significance of the research direction can be highlighted. The overfitting issue seems to occur in specific kinds of DIP models. Recent diffusion models have outstanding performance on zero-shot image restoration tasks without the overfitting issue. With the fixed interactions, the diffusion model is able to denoise images iteratively. Thus, for the diffusion models, the ES strategy may be not necessary.
>
> We respectfully disagree with the reviewer’s comments from three angles:
>
> * DIP has totally different settings and niche areas than zero-shot diffusion-based models. Zero-shot diffusion-based methods still need pre-trained models based on large-scale training sets while DIP does not need any training data or pre-trained models. Hence, the DIP models are invaluable for inverse problems where collecting sufficient data to pre-train a diffusion model is prohibitive or infeasible, e.g., in microscopy imaging of rare scientific objects.
> * We agree that diffusion models are very powerful generative models, but most of the existing diffusion-based models only deal with linear image restoration tasks. We could not find much work on zero-shot diffusion-based methods for solving nonlinear image restoration tasks (e.g. blind image deblurring (BID)). However, DIP-based models have already made breakthroughs in the BID task—for which collecting large-scale training sets in practice is challenging [1, 2], and obtained even better results than state-of-the-art data-driven methods [3,4].
> * Diffusion models can also have overfitting issues, especially when the observation is noisy. In the new Section 3.8, we confirm this on zero-shot image super-resolution based on diffusion models from [5]. The overfitting issue is evident with or without additional noise; it is unclear how they set their fixed number of iterations, but they yield suboptimal results. The preliminary test suggests that our ES-WMV can detect near-peak performance, and return comparable or better results than theirs. We leave mitigating the overfitting issue of diffusion models for future work since it is out of the scope of this paper, where our focus is on DIP models.
>
>
> ### Q2. The relationship between the proposed ES strategy and the theorems is not clear. I do not check the correctness of the theorems. It would be better to provide some intuitive understanding of these theorems. Is Theorem 2.1 suitable for most models, e.g., diffusion models? The authors can put the important theorems in the main paper, and leave other theorems in the supplementary materials.
>
> * Our Theorems 2.1 and 2.2 provide a lower and an upper bound, respectively, for the VAR curve, as shown in Figure 5. The success of our ES methods relies on (1) the U-shape of the VAR curve,  (2) the VAR valley aligns with the PSNR peak, and (3) VAR valley can be numerically detected. Our theory mostly sheds light on (1). The theory is based on the popular neural tangent kernel idea to gain insights into deep learning, which linearizes the training behavior around the initialization. As we acknowledge in the paper, the theory is still very loose and limited, as a first step toward understanding. The nature of this paper is mostly empirical.
>
> * Theorems 2.1 and 2.2 are for DIP only, under specific conditions on the architecture (2-layer) and the problem (additive denoising). It does not cover diffusion models.
>
> * We have already put the important theorems in the main paper and left proofs in the Appendix.
>
>
> ### Q3. The authors show extensive examples of how to detect the ES points. It is also important to provide more visual comparisons in the experiments.
>
> * Thanks for the suggestion! We have added more visual comparisons in Figures 7 and 12.

---

> ### Author Response · Authors · 2023-03-30
> **Response to Reviewer XgzK (2/2)**
>
> ### Q4. It would be better to compare with some SOTA methods. Besides, the authors can move the super-resolution results in the main paper.
>
> * Thank you for the suggestion about comparison! However, it is NOT our intention to claim or show the original DIP and DIP+our ES (DIP+ES) methods can produce SOTA results on any particular inverse problem. On simple inverse problems such as image denoising, image inpainting, and super-resolution, DIP is not always as competitive as SOTA data-driven methods as shown even in the original DIP paper. While on complicated nonlinear inverse problems, DIP has obtained better results than SOTA data-driven methods, e.g., on blind image deblurring [3, 4]. So, whether to choose DIP or not is problem-dependent, and not our focus here.
>
> * Our focus is to solve the early-learning-then-overfitting issue in DIP when used to solve visual inverse problems, and our main contribution is proposing and validating an effective, efficient, general, and robust ES method that can locate an ES point resulting in near-peak performance with respect to the original. Hence, while the suggested comparison is definitely valuable if the goal is to derive the best method for any specific inverse problem, it may be confusing and misleading here—we are only concerned with the gap between DIP and our DIP+ES.
>
> * We have moved sections on image inpainting and super-resolution to the new Section 3.2 and 3.3, respectively.
>
> > [1] Zhang, K., Ren, W., Luo, W., Lai, W.S., Stenger, B., Yang, M.H. and Li, H., 2022. Deep image deblurring: A survey. International Journal of Computer Vision, 130(9), pp.2103-2130.
> >
> > [2] Koh, J., Lee, J. and Yoon, S., 2021. Single-image deblurring with neural networks: A comparative survey. Computer Vision and Image Understanding, 203, p.103134.
> >
> > [3] Ren, D., Zhang, K., Wang, Q., Hu, Q. and Zuo, W., 2020. Neural blind deconvolution using deep priors. In Proceedings of the IEEE/CVF Conference on Computer Vision and Pattern Recognition (pp. 3341-3350).
> >
> > [4] Zhuang, Z., Li, T., Wang, H. and Sun, J., 2022. Blind Image Deblurring with Unknown Kernel Size and Substantial Noise. arXiv preprint arXiv:2208.09483.
> >
> > [5] Wang, Y., Yu, J. and Zhang, J., 2022. Zero-Shot Image Restoration Using Denoising Diffusion Null-Space Model. arXiv preprint arXiv:2212.00490.

---

### Review · Reviewer_6brH · 2023-03-06

**Summary Of Contributions:**

The paper proposes a new criterion for early stopping, which is extremely simple and intuitive, and also barely requires additional computational resources. However, such methods will inevitably be sensitive to many different settings, and there seems a deficiency of discussions in this regard. Overall, I think this is a good paper, but some additional discussions need to be addressed clearly.

**Audience:**

Yes

**Broader Impact Concerns:**

none noted

**Claims And Evidence:**

Yes

**Requested Changes:**

My biggest concern is that this method will have limitations as agreed upon by the authors. While it is fine to have limitations, the paper should present the pros and cons of this new proposed method much more cleanly to guide future users to use this method, more specifically,
   - Based on the intuition of the method, the effectiveness of the method will depend on configurations such as step sizes, fluctuations of the curves etc. So far, the current experiment only shows certain "regular" cases where the step sizes are sufficiently small, and has no discussions on these cases.
   - While the authors have discussed various comparisons of performances of the proposed method with respect to others, and list pros and cons, the discussions are shattered across the manuscript and it seems hard to have an overall picture of when the proposed method works and when not.
        - I will recommend the authors summarize these results into a table, under different tasks, noise levels, different configurations, maybe computing concerns, which scenarios the proposed method is preferred.

**Strengths And Weaknesses:**

- strengths
   - the method proposed is extremely simple, and intuitive, and, according to the presented results, have superior performances
   - the written of the paper is clear and easy to follow
- weakness
   - I will suggest a much deeper discussion of this method over different settings in comparison to other methods, and especially a summary of such comparisons (see below).

---

> ### Author Response · Authors · 2023-03-30
> **Response to Reviewer 6brH**
>
> We thank reviewer 6brH for the detailed review and insightful comments.
>
> ### Q1. Based on the intuition of the method, the effectiveness of the method will depend on configurations such as step sizes, fluctuations of the curves etc. So far, the current experiment only shows certain "regular" cases where the step sizes are sufficiently small, and has no discussions on these cases.
>
> * We are a bit confused by the reviewer’s comment, particularly “... on these cases”. Is it on “on other cases”? We hope the reviewer can help to clarify the question. If the question is about other learning rates, in the new Figure 14, we compare the PSNR and VAR curves with default and smaller learning rates—we used the default learning rates in the previous version of the paper, respectively (see also the new Figure 16 that contains updated results with smaller learning rates). Large learning rates can lead to fluctuating PSNR curves, affecting the detection performance of our ES methods, as can be see from the previous version of the paper. In general, the step size should be sufficiently small to ensure the PSNR and VAR curves are sufficiently smooth to allow effective ES detection.
>
>
> ### Q2. I will recommend the authors summarize these results into a table, under different tasks, noise levels, different configurations, maybe computing concerns, which scenarios the proposed method is preferred.
>
> * Thanks for the suggestion! We have summarized the results in the new Table 1.

---

### Review · Reviewer_3j1D · 2023-03-15

**Summary Of Contributions:**

This work proposes an early stopping mechanism for DIP based on the windowed moving variance of the output images. An early stop is detected when the variance does not decrease after a certain number of training steps. The proposed method is observed to have small PSNR and SSIM gaps compared to other early stopping methods and is applicable to different DIP variants.

**Audience:**

Yes

**Claims And Evidence:**

Yes

**Requested Changes:**

See the weakness part.

**Strengths And Weaknesses:**

Strengths:

1. The proposed method is simple enough and to some extent aligned with the reconstruction quality. Theoretical analysis is also provided.
2. The paper is well-written with a lot of informative details. Extensive experiments do show the superiority of the the method. And the proposed method is generally applicable to various types of noises.
3. Interesting discussions are provided, supported by observations in multiple use cases. The limitation is also discussed in different scenarios.

Weaknesses:

1. What I am pretty concerned about is the usability of the proposed ES method considering there are many failure cases. The most important thing is that there is no such a wholistic understanding of on what kind of input images the failure would happen. This could be fatal in the application considered in this paper where there are no reference images during inference. The failure cases shown in the last page show multi-peak curves and extreme fluctuations in PSNR during training. This should be investigated more closely. The authors mention that they "observe that using smaller learning rates for GP-DIP and DD helps to smooth out their curves and mitigate the multi-valley phenomenon, which likely will lead to much smaller detection gaps. We hesitate to refine in this direction, as our focus of this paper is on the ES method itself.", which I humbly disagree. This is critical to the usability the ES method. Will the performance (in terms of PSNR for example) downgrade with a smaller learning rate? If a smaller learning rate is universally in every aspect, like smoothening the curves and getting better PSNR, why do not the authors just report results of experiments with smaller learning rate? This should be discussed with more details.

---

> ### Author Response · Authors · 2023-03-30
> **Response to Reviewer 3j1D**
>
> We thank reviewer 3j1D for the detailed review and insightful comments.
>
> ### Q1. What I am pretty concerned about is the usability of the proposed ES method considering there are many failure cases. The most important thing is that there is no such a wholistic understanding of on what kind of input images the failure would happen.
> * In the revision, we provide more detailed statistics of failure cases. In particular, for the large-scale real-world denoising reported in Table 2 (only mean detection gaps are reported in the original version), we include the histogram of the gap distribution (Figure 27): for 95% of the images, the PSNR gap is below 2dB. For Figures 11 and 16 where there were 5-6 cases with PSNR gaps above 2dB, we are able to substantially improve the results by reducing the learning rates, i.e., with only 2-3 cases with >=2 PSNR gaps.
>
> * Our ES method needs three things to succeed: (1) the U-shape of the VAR curve,  (2) the VAR valley aligning with the PSNR peak, and (3) successful numerical detection of the VAR valley. In the new Section 3.6, we discuss two major failure modes of our method: (I) the VAR valley aligns well with the PSNR peak, but the U-shape assumption is violated. A dominant pattern is the presence of multiple valleys; (II) the VAR valley does not align well with the PSNR peak, which often happens on images with significant high-frequency components. More details can be found in Section 3.6.
>
>
> ### Q2. If a smaller learning rate is universally in every aspect, like smoothening the curves and getting better PSNR, why do not the authors just report results of experiments with smaller learning rate?
>
> * Thanks for raising the point! For GP-DIP and DD that concern this smaller learning comment, our original experiment used their default hyperparameters, including default learning rates. We were a bit hesitant to finetune their hyperparameters to lead to any bias. Per the reviewer’s comment, we redo the experiments for GP-DIP and DD with smaller learning rates and include the results in Figures 14 and 16. It is clear smaller learning rates smooth out the PSNR and VAR curve and hence lead to smaller detection gaps.

---

### Author Response · Authors · 2023-03-30
**General Response to All Reviewers**

We thank the reviewers for their thoughtful and constructive comments about our manuscript! We have carefully revised our submission accordingly and highlighted the major changes in **blue**. Below, we first summarize the major changes and then address the reviewers’ individual comments.

* We have added Table 1 to summarize the performance comparison between our ES-WMV method and other competing methods on all the major tasks we experiment with;
* We have added Section 3.8 to show the overfitting issue of diffusion-based models when applied to super-resolution and the potential of our ES-WMV method in mitigating this issue—since the focus of this paper is to perform ES for DIP-based methods, we leave a systematic study of this to future work;
* We have added Figure 27 which visualizes the distribution of detection gaps on real-world denoising (mean detection gaps are reported in Table 2). It shows there is only a small portion of failure cases among the 1024 images we test;
* We have added Section 3.6 to discuss the major failure modes;
* We have added Figures 14 and 16 for DD and GP-DIP to show that smaller learning rates help to smooth out PSNR curves and consequently decrease detection gaps;
* We have moved both image inpainting and super-resolution to Section 3.2 and 3.3.
* We have added more visual results in Figures 7 and 12.

---

### Decision · Action_Editors · 2023-04-25

**Recommendation:** Reject

**Comment:**

This manuscript explores a novel research direction, deep image prior, and its early-stopping regularization strategy for solving inverse problems in computational imaging. The proposed strategy is shown to be effective and operationally usable for achieving near-peak performance without ground truth access.

The AE acknowledges the paper's new research effort and finds the proposed early-stopping strategy to be both operationally simple and well-justified. Nonetheless, the reviews have pointed out that the experimental evaluation is inadequate. While the proposed approach is applied to image processing tasks, including image super-resolution, it would be beneficial to compare its performance with state-of-the-art techniques such as diffusion models. In addition, regarding image super-resolution, the manuscript lacks visual comparisons of the results among all the super-solvers.

The AE suggests a Major Revision for this manuscript and encourages the authors to carefully address the reviewers' comments and make significant revisions before resubmitting their work to TMLR.



**Audience:**

Yes

**Claims And Evidence:**

Not all.

While the theoretical evidence appears sufficient, the experimental evidence is currently insufficient.

While the proposed approach is applied to image processing tasks, including image super-resolution, it would be beneficial to compare its performance with state-of-the-art techniques such as diffusion models. In addition, regarding image super-resolution, the manuscript lacks visual comparisons of the results among all the super-solvers.